# Coarse-Grained Boltzmann Generators

Weilong Chen [* 1]   Bojun Zhao [* 1]   Jan Eckwert [1]   Julija Zavadlav [1 2]

## Abstract

Sampling equilibrium molecular configurations from the Boltzmann distribution is a longstanding challenge. Boltzmann Generators (BGs) address this by combining exact-likelihood generative models with importance sampling, but practical scalability is limited. Meanwhile, coarse-grained surrogates enable the modeling of larger systems by reducing effective dimensionality, yet often lack a reweighting procedure required to ensure asymptotically correct statistics. In this work, we propose Coarse-Grained Boltzmann Generators (CG-BGs), a framework for reduced-order generative modeling with importance sampling in coarse-grained coordinate space. CG-BGs generate samples using a flow-based model and reweight them using a learned potential of mean force (PMF). We show that the PMF can be learned from rapidly converged trajectories via enhanced sampling force matching. Experiments demonstrate that CG-BGs capture solvent-mediated interactions in highly reduced representations while substantially reducing computational cost relative to atomistic BGs, providing a practical route toward equilibrium sampling of larger molecular systems.

## 1. Introduction

Accurately sampling molecular configurations from the Boltzmann distribution is a central problem in statistical physics (Chandler, 1987). These samples are required for estimating observables and thermodynamic quantities, including free energies (Chipot & Pohorille, 2007). For molecular systems, the high dimensionality of the configuration space makes direct computation of the partition function intractable, forcing a reliance on simulation methods such as Molecular Dynamics (MD) or Markov Chain Monte Carlo (MCMC) (Frenkel & Smit, 2002). However, these methods are often inefficient in systems with rugged energy landscapes. High free energy barriers lead to metastable trapping, producing strongly correlated samples and slow convergence (Lindorff-Larsen et al., 2011). Despite extensive progress in enhanced sampling methods (Zhu et al., 2025; Hénin et al., 2022) (e.g., umbrella sampling (Torrie & Valleau, 1977), metadynamics (Laio & Gervasio, 2008)) and coarse-graining (Noid, 2013; Saunders & Voth, 2013), equilibrium sampling at scale remains difficult.

Deep generative models have recently emerged as a promising alternative for equilibrium sampling (Noé et al., 2019; Albergo et al., 2019; Wirnsberger et al., 2020). A notable example is the Boltzmann Generator (BG) (Noé et al., 2019). By learning a diffeomorphic transformation between a simple prior (e.g., a Gaussian) and the complex molecular configuration space, BGs enable efficient proposal generation and exact-likelihood evaluation. The tractable likelihood permits rigorous reweighting of generated configurations to the target Boltzmann distribution via importance sampling, allowing unbiased estimation of equilibrium observables. This has enabled amortized sampling strategies (Tan et al., 2025b), where generation is substantially cheaper than MD simulations.

In practice, however, scaling BGs to high-dimensional molecular systems remains difficult (Klein & Noé, 2024; Tan et al., 2025a). As system size increases, even expressive generative models exhibit diminishing overlap with the target distribution, resulting in high-variance importance weights and ineffective reweighting. In addition, likelihood evaluation requires Jacobian determinant computation (Chen et al., 2018), which introduces significant computational overhead and scales poorly with dimensionality (Hutchinson, 1989). As a result, existing BG applications are largely restricted to small systems such as short peptides (Tan et al., 2025b; Rehman et al., 2025) with empirical implicit solvent models (Hawkins et al., 1995; Nguyen et al., 2013).

Coarse-graining (CG) provides a complementary approach by projecting atomistic configurations onto a lower-dimensional set of collective variables. This idea underlies

---
[*]Equal contribution, random order. [1]Professorship of Multiscale Modeling of Fluid Materials, Department of Engineering Physics and Computation, TUM School of Engineering and Design, Technical University of Munich, Germany [2]Atomistic Modeling Center, Munich Data Science Institute, Technical University of Munich, Germany. Correspondence to: Julija Zavadlav <julija.zavadlav@tum.de>.

*Proceedings of the 43rd International Conference on Machine Learning*, Seoul, South Korea. PMLR 306, 2026. Copyright 2026 by the author(s).

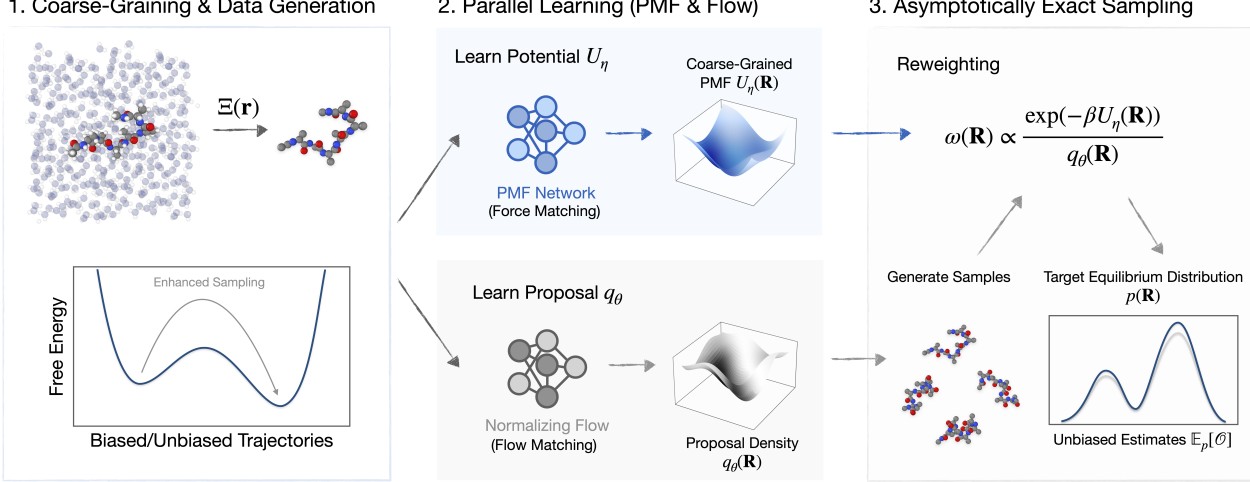

*Figure 1.* **CG-BG framework.** Atomistic configurations are mapped to CG coordinates to construct training data. A PMF network learns $U_\eta(\mathbf{R})$ from rapidly converged data, while a normalizing flow learns a proposal density $q_\theta(\mathbf{R})$ over CG configurations. Samples generated from the flow are reweighted using the PMF to recover the target equilibrium distribution $p(\mathbf{R})$, enabling unbiased estimation of thermodynamic observables.

Boltzmann Emulators (Lewis et al., 2025; Zheng et al., 2024; Jing et al., 2024; Zhu et al., 2026) and related generative surrogates (Schreiner et al., 2023; Daigavane et al., 2025; Plainer et al., 2025; Costa et al., 2025; dos Santos Costa et al., 2024; Diez et al., 2025; Vlachas et al., 2021; Xu et al., 2025). By reducing the number of degrees of freedom during generation, these methods can be applied to larger systems. However, training requires converged unbiased simulation data, which is difficult to obtain in practice. Thus, CG models are often trained on short finite-time trajectories that do not fully capture the target distribution (Lewis et al., 2025; Zheng et al., 2024) and, unlike Boltzmann Generators, do not incorporate reweighting to correct this mismatch due to the lack of an explicit energy function for the target distribution, resulting in biased statistical estimates.

**Present work**. In this paper, we introduce *Coarse-Grained* Boltzmann Generators (CG-BGs, Fig. 1), a class of Boltzmann Generators that operate directly in CG coordinates [1]. CG-BGs combine generative modeling with importance sampling using a learned potential of mean force (PMF) as the target energy, enabling asymptotically correct equilibrium sampling in a reduced-dimensional representation. This design provides a scalable pathway for sampling high-dimensional molecular systems. CG-BGs are particularly advantageous as they can be trained directly from rapidly converged data and effectively capture complex solvent-mediated effects.

Our main contributions are:

- We introduce *Coarse-Grained* **Boltzmann Generators**, a scalable framework for equilibrium sampling in CG coordinate space using **machine learning potentials** (MLPs) as the target energy for importance sampling.

- We show that enhanced sampling force matching enables learning the PMF from rapidly converged simulation trajectories, eliminating reliance on unbiased equilibrium data and providing a correction mechanism for Boltzmann Emulators.

- We demonstrate that CG-BGs capture solvent-mediated interactions in highly reduced representations, achieving improved accuracy over classical implicit solvent models while substantially reducing computational cost relative to atomistic BGs.

## 2. Background and Preliminaries

**Notation**. We use lowercase variables for fine-grained (atomistic) quantities and uppercase variables for CG quantities.

We consider a many-body system with configuration $\mathbf{r} \in \mathbb{R}^n$ governed by a potential energy function $u(\mathbf{r})$. At thermodynamic equilibrium with temperature $T$, the system follows the Boltzmann distribution

$$p(\mathbf{r}) = \frac{e^{-\beta u(\mathbf{r})}}{Z}, \quad Z = \int e^{-\beta u(\mathbf{r})} d\mathbf{r}, \qquad (1)$$

where $\beta = (k_B T)^{-1}$ and $Z$ is the partition function. The goal of equilibrium sampling is to generate samples from $p(\mathbf{r})$ in order to compute observables $\mathbb{E}_p[\mathcal{O}] =$

---

[1] Our code is publicly available at https://github.com/tummfm/cg-bg.

$\int \mathcal{O}(\mathbf{r})p(\mathbf{r})d\mathbf{r}$, and free energies. A dataset is considered converged if empirical averages of observables match their equilibrium expectations within statistical error.

## 2.1. Boltzmann Generators and Emulators

Boltzmann Generators (BGs) (Noé et al., 2019) combine exact-likelihood generative models with importance sampling to estimate equilibrium properties. Typically implemented using normalizing flows (Rezende & Mohamed, 2016), a BG defines a tractable proposal density $q_\theta(\mathbf{r})$ that approximates $p(\mathbf{r})$. Given samples $\mathbf{r}_i \sim q_\theta$, importance weights are computed as

$$w(\mathbf{r}_i) = \frac{p(\mathbf{r}_i)}{q_\theta(\mathbf{r}_i)} \propto \frac{e^{-\beta u(\mathbf{r}_i)}}{q_\theta(\mathbf{r}_i)}. \quad (2)$$

Unbiased estimates of equilibrium expectations are then obtained using the self-normalized importance sampling estimator

$$\mathbb{E}_p[\mathcal{O}] \approx \frac{\sum_{i=1}^N w(\mathbf{r}_i)\,\mathcal{O}(\mathbf{r}_i)}{\sum_{i=1}^N w(\mathbf{r}_i)}. \quad (3)$$

Provided that $q_\theta$ overlaps sufficiently with $p$, this estimator converges to the corresponding Boltzmann averages. This allows BGs to be trained on biased or non-equilibrium samples, with reweighting correcting the induced distribution shift at evaluation.

Boltzmann Emulators adopt similar generative architectures but omit the reweighting step, relying directly on $q_\theta$ for estimating observables. Model accuracy is therefore determined by the quality of the learned distribution $q_\theta$, with no correction applied at inference time. This places stronger requirements on the training data: accurate models require long unbiased trajectories, which are difficult to obtain in practice.

## 2.2. Continuous Normalizing Flows

Continuous normalizing flows (CNFs) extend discrete normalizing flows (Dinh et al., 2014; Rezende & Mohamed, 2016) by modeling density transformations as solutions to time-dependent ordinary differential equations (ODEs) (Chen et al., 2018). A vector field $v_\theta : [0,1] \times \mathbb{R}^n \to \mathbb{R}^n$, parameterized by a neural network, defines the dynamics

$$\frac{d\mathbf{x}(t)}{dt} = v_\theta(t, \mathbf{x}(t)), \qquad \mathbf{x}(0) \sim p_0, \quad (4)$$

where $p_0$ is a simple prior distribution. The solution at time $t$ is

$$\mathbf{x}(t) = \mathbf{x}(0) + \int_0^t v_\theta(\tau, \mathbf{x}(\tau))\, d\tau. \quad (5)$$

The evolution of the log-density follows the instantaneous change-of-variables formula

$$\log p_t(\mathbf{x}(t)) = \log p_0(\mathbf{x}(0)) - \int_0^t \nabla \cdot v_\theta(\tau, \mathbf{x}(\tau))\, d\tau, \quad (6)$$

where $\nabla \cdot v_\theta$ denotes the divergence of the vector field.

While CNFs are often trained using maximum likelihood, Flow Matching (FM) (Lipman et al., 2022; Liu et al., 2022; Albergo et al., 2023) provides a simulation-free alternative. Conditional Flow Matching (CFM) (Tong et al., 2023) directly regresses the neural vector field $v_\theta$ onto a target conditional vector field $u_t(\mathbf{x} \mid z)$ that induces a prescribed probability path. The training objective is

$$\mathcal{L}_{\text{CFM}}(\theta) = \mathbb{E}_{t,z,\mathbf{x}\sim p_t(\cdot|z)}\big[\|v_\theta(t,\mathbf{x}) - u_t(\mathbf{x} \mid z)\|^2\big], \quad (7)$$

where $t \sim \mathcal{U}[0,1]$ and $z$ is a conditioning variable. A common choice uses linear interpolation between paired source and target samples (Albergo et al., 2023; Tong et al., 2023). Let $z = (\mathbf{x}_0, \mathbf{x}_1)$ with $\mathbf{x}_0$ and $\mathbf{x}_1$ sampled from the source and target distributions, respectively. The interpolated state and corresponding vector field are

$$\mathbf{x}_t = (1-t)\mathbf{x}_0 + t\mathbf{x}_1, \qquad u_t(\mathbf{x}_t \mid \mathbf{x}_0, \mathbf{x}_1) = \mathbf{x}_1 - \mathbf{x}_0. \quad (8)$$

## 2.3. Coarse-Graining and Potentials of Mean Force

Coarse-graining maps atomistic configurations $\mathbf{r} \in \mathbb{R}^n$ to a lower-dimensional set of collective variables (CVs), or *beads*, $\mathbf{R} \in \mathbb{R}^N$ with $N \ll n$, through a mapping $\mathbf{R} = \Xi(\mathbf{r})$. The CG variables are typically chosen to retain the slow degrees of freedom. In *bottom-up* coarse-graining (Noid et al., 2008; Jin et al., 2022), the objective is to construct an effective CG potential such that the CG model reproduces the marginal equilibrium distribution of the atomistic system:

$$p(\mathbf{R}) = \int p(\mathbf{r})\,\delta(\Xi(\mathbf{r}) - \mathbf{R})\, d\mathbf{r}. \quad (9)$$

This marginal distribution admits a Boltzmann form,

$$p(\mathbf{R}) \propto e^{-\beta U(\mathbf{R})}, \quad (10)$$

where the effective energy $U(\mathbf{R})$, known as the *potential of mean force*, is defined up to an additive constant as

$$U(\mathbf{R}) = -k_B T \ln \int e^{-\beta u(\mathbf{r})}\,\delta(\Xi(\mathbf{r}) - \mathbf{R})\, d\mathbf{r}. \quad (11)$$

The PMF includes both energetic and entropic contributions from the eliminated degrees of freedom and generally contains many-body, state-dependent interactions (Krishna et al., 2009).

Evaluating the PMF is intractable in practice due to the high-dimensional integral over $\mathbf{r}$. Classical CG force fields (Marrink et al., 2007; Souza et al., 2021) approximate $U(\mathbf{R})$ using fixed functional forms, which may lack sufficient expressivity. More recent approaches represent $U(\mathbf{R})$ using neural networks trained by force matching (Noid et al., 2008; Wang et al., 2019) or relative entropy minimization (Shell, 2008; Thaler et al., 2022).

# 3. Coarse-Grained Boltzmann Generators

Atomistic BGs permit reweighting-based equilibrium sampling but become difficult to apply to large systems. Boltzmann Emulators improve scalability by omitting reweighting, at the cost of introducing bias.

We introduce CG-BGs, which perform generative modeling and importance sampling directly in CG coordinate space. Instead of the full atomistic distribution, CG-BGs target the marginal distribution $p(\mathbf{R})$ defined by the PMF.

CG-BGs consist of two components: a flow-based model that generates CG configurations and a learned PMF used for importance reweighting. Unlike atomistic MLPs trained on labeled energies (Blank et al., 1995; Behler & Parrinello, 2007), CG PMFs cannot be directly evaluated from atomistic configurations because they include entropic contributions from eliminated degrees of freedom. We next describe how the PMF is learned from atomistic simulations and outline the CG-BG workflow (Fig. 1).

## 3.1. Variational Force Matching

Variational Force Matching (VFM) (Noid et al., 2008), also known as multiscale coarse-graining, is a bottom-up approach for learning the PMF from atomistic forces.

The central condition is that CG forces should match, in expectation, the instantaneous atomistic forces projected onto the CG coordinates, denoted by $\mathcal{F}_{\text{proj}}(\mathbf{r})$. The exact PMF satisfies:

$$-\nabla U(\mathbf{R}) = \mathbb{E}_{p(\mathbf{r}|\mathbf{R})}\big[\mathcal{F}_{\text{proj}}(\mathbf{r})\big]. \tag{12}$$

where the expectation is taken over the *fiber distribution* (Hummerich et al., 2025), i.e., the conditional distribution of atomistic configuration $\mathbf{r}$ given $\mathbf{R}$.

From a learning perspective, instantaneous projected forces provide stochastic estimates of the conditional mean force in Eq. (12). Given a dataset $\mathcal{D}$ of atomistic configurations, a parameterized CG potential $U_\eta(\mathbf{R})$ is trained by minimizing

$$\mathcal{L}_{\text{VFM}}(\eta) = \mathbb{E}_{\mathbf{r}\sim\mathcal{D}}\Big[\big\|\nabla_{\mathbf{R}}U_\eta(\Xi(\mathbf{r})) + \mathcal{F}_{\text{proj}}(\mathbf{r})\big\|_2^2\Big]. \tag{13}$$

When $\mathcal{D}$ is sampled from equilibrium, this objective minimizes the Fisher divergence between the model distribution $p_\eta(\mathbf{R})$ and the true marginal $p(\mathbf{R})$. Theoretical error bounds follow from Log-Sobolev inequalities (Proof in §A.1):

**Proposition 1.** *Let* $p^*(\mathbf{R}) \propto e^{-\beta U^*(\mathbf{R})}$ *be the true marginal and* $p_\eta(\mathbf{R}) \propto e^{-\beta U_\eta(\mathbf{R})}$ *the learned distribution. If* $p^*$ *satisfies a Logarithmic Sobolev Inequality (LSI) with constant* $\rho > 0$. *Then, the Kullback-Leibler divergence between the learned and true distributions*

*is bounded by the expected squared force error:*

$$\mathcal{D}_{\text{KL}}(p_\eta\|p^*) \leq \frac{\beta^2}{2\rho}\mathbb{E}_{p_\eta}\big[\|\nabla U_\eta(\mathbf{R}) - \nabla U^*(\mathbf{R})\|^2\big]. \tag{14}$$

While global LSI conditions are strong assumptions for multimodal PMFs (Vempala & Wibisono, 2019), this result motivates force matching as a proxy for distributional accuracy. This is relevant for importance sampling, where performance depends on the divergence between the learned and true marginal distributions.

## 3.2. Enhanced Sampling for Force Matching

Standard force matching requires unbiased converged data, which is expensive to obtain for systems with metastable states and large free energy barriers—a limitation shared by Boltzmann Emulators. In addition, high-energy transition regions are rarely visited under the Boltzmann distribution, yet are important for accurately learning the PMF.

Enhanced sampling force matching (ESFM) (Chen et al., 2026) overcomes these limitations by using invariance of the fiber distribution under coarse-grained biasing.

**Proposition 2.** (Chen et al. (2026)) *Let* $V(\mathbf{R})$ *be a bias potential depending only on the coarse-grained coordinates. The conditional distribution of atomistic configurations given* $\mathbf{R}$ *is invariant:*

$$p_V(\mathbf{r} \mid \mathbf{R}) = p(\mathbf{r} \mid \mathbf{R}). \tag{15}$$

Since the mean force $-\nabla U(\mathbf{R})$ is an expectation of the projected forces over this conditional distribution (Eq. 12), the regression target is unchanged under CG biasing. ESFM minimizes

$$\mathcal{L}_{\text{ESFM}}(\eta) = \mathbb{E}_{\mathbf{r}\sim\mathcal{D}_{\text{bias}}}\Big[\big\|\nabla_{\mathbf{R}}U_\eta(\Xi(\mathbf{r})) + \mathcal{F}_{\text{proj}}(\mathbf{r})\big\|_2^2\Big], \tag{16}$$

where $\mathcal{D}_{\text{bias}}$ is a rapidly converged dataset generated using enhanced sampling and $\mathcal{F}_{\text{proj}}(\mathbf{r})$ denotes forces recomputed from the unbiased atomistic potential.

**Proposition 3.** (Chen et al. (2026)) *Minimizing* $\mathcal{L}_{\text{ESFM}}$ *yields the same global optimum as standard force matching loss* $\mathcal{L}_{\text{VFM}}$, *assuming sufficient model expressivity.*

Together, Propositions 2 (Proof in §A.2) and 3 establish that ESFM enables accurate PMF learning from biased enhanced sampling data, benefiting from faster convergence and improved coverage of transition regions.

## 3.3. The CG-BG Workflow

After training, the learned PMF $U_\eta$ defines the target energy for importance sampling, rather than as a force field for MD

integration. Let $q_\theta(\mathbf{R})$ denote the density induced by the trained flow model. Importance weights are computed as

$$w(\mathbf{R}) \propto \frac{\exp(-\beta U_\eta(\mathbf{R}))}{q_\theta(\mathbf{R})}. \qquad (17)$$

Provided $U_\eta$ accurately approximates the true PMF on the support of $q_\theta$, reweighting gives unbiased estimates under $p(\mathbf{R})$ in the idealized setting where the PMF is exact. In practice, the learned PMF introduces approximation bias.

The reliability of importance reweighting is quantified by the normalized effective sample size (ESS) (Kish, 1965),

$$\text{ESS} = \frac{1}{B} \frac{\left(\sum_{i=1}^{B} w(\mathbf{R}_i)\right)^2}{\sum_{i=1}^{B} w(\mathbf{R}_i)^2}. \qquad (18)$$

where $B$ denotes the number of generated samples. The normalized ESS takes values in $(0, 1]$, with larger values indicating better overlap between the generative density $q_\theta$ and the target density defined by $U_\eta$. In practice, machine learning potentials may exhibit unphysical extrapolation outside the training domain, and generative models may occasionally produce high-energy artifacts, both of which can lead to weight degeneracy. To improve robustness, we apply a weight clipping strategy (Tan et al., 2025a; Gloy & Olsson, 2025; Moqvist et al., 2025) to truncate statistical outliers before computing expectations (See §G.4). The complete CG-BG training and sampling pipeline is summarized in Fig. 1.

## 4. Experiments

We evaluate CG-BGs on the Müller–Brown (MB) potential and three alanine peptide systems, including alanine dipeptide (Ac-Ala-NHMe, 22 atoms), alanine tripeptide (Ac-Ala$_3$-NHMe, 42 atoms), and alanine hexapeptide (Ac-Ala$_6$-NHMe, 72 atoms). Additional experimental details including architectures are provided in §C and §D. CG-BG samples are generated and reweighted following the algorithms described in §F.

**Datasets**. For all systems, we construct unbiased and biased datasets for training and evaluation. Biased datasets are generated using enhanced sampling methods to accelerate exploration. For the MB system, data are generated via Langevin dynamics (§B.1), with umbrella sampling (Torrie & Valleau, 1977) used in the biased setting to improve transitions between metastable basins. For peptide systems, datasets are generated using both explicit and implicit solvent models (§B.2). Explicit solvent data are produced using a classical force field (Lindorff-Larsen et al., 2010) and include long unbiased MD trajectories as well as biased simulations obtained via well-tempered metadynamics (Barducci et al., 2008). Implicit solvent data are generated using the same force field in combination with a generalized

Born model under different parameterizations (OBC1 and OBC2) (Onufriev et al., 2004). For explicit solvent simulations, configurations are further coarse-grained using either a *Heavy Atom* mapping (Fig. 2a and Fig. 4a), which retains all heavy atoms, or a *Core Beta* mapping (Fig. 2f and Fig. 4f), which retains backbone atoms and the $C_\beta$ position. Full details on dataset generation and enhanced sampling procedures are provided in §B.

**Baselines**. Unlike previous BG work (Klein & Noé, 2024; Tan et al., 2025a;b), which often treats implicit solvent simulations as reference, we use explicit solvent simulations as the primary reference and treat empirical implicit solvent models as baselines. We additionally report results from atomistic BGs, including TarFlow and ECNF++ (Tan et al., 2025b) trained on implicit solvent simulation data.

**Metrics**. We report ESS, Jensen-Shannon (JS) divergence, and PMF error. JS divergence is computed between the sampled and reference dihedral angle free energy profiles. The PMF error is defined as the squared distance between the negative logarithms of the sampled and reference densities, placing additional emphasis on low-probability regions compared to JS divergence (Plainer et al., 2025; Durumeric et al., 2024). Energy histograms and free energy profiles of $\phi$ dihedral are shown in the main text, while additional results, including $\psi$ dihedral free energy profiles, Ramachandran plots (Ramachandran, 1963), bond length distributions, and weight clipping ablations, are provided in §G.

### 4.1. Recovering Equilibrium Distributions from Biased and Unbiased Data

We first demonstrate that CG-BGs inherit the importance reweighting capability of atomistic BGs, enabling recovery of equilibrium statistics from flow models trained on either biased or unbiased trajectories.

For the MB system (Fig. 6), coarse-graining corresponds to projection onto the $x$-coordinate, yielding an analytically exact reference $U(x) = -k_B T \ln \int \exp\left(-\beta u(x, y)\right) dy$ (Fig. 6c–d). For peptides, we use two CG mappings depending on system size: the Heavy Atom mapping for alanine dipeptide and tripeptide (Fig. 2a and Fig. 4a), and the Core Beta mapping for hexapeptide (Fig. 4f). Throughout §4.1 and §4.2, the PMF is learned from rapidly converged datasets using ESFM (§3.2).

As shown in Fig. 2c, Fig. 4c, 4h and Fig. 6c, despite training on long unbiased datasets and using expressive models, the raw flow proposals deviate from the MD reference. These discrepancies arise from low-quality samples generated by the flow model, a limitation intrinsic to Boltzmann Emulators that cannot be systematically corrected without reweighting. After reweighting, CG-BGs successfully recover equilibrium free energy profiles in close agreement

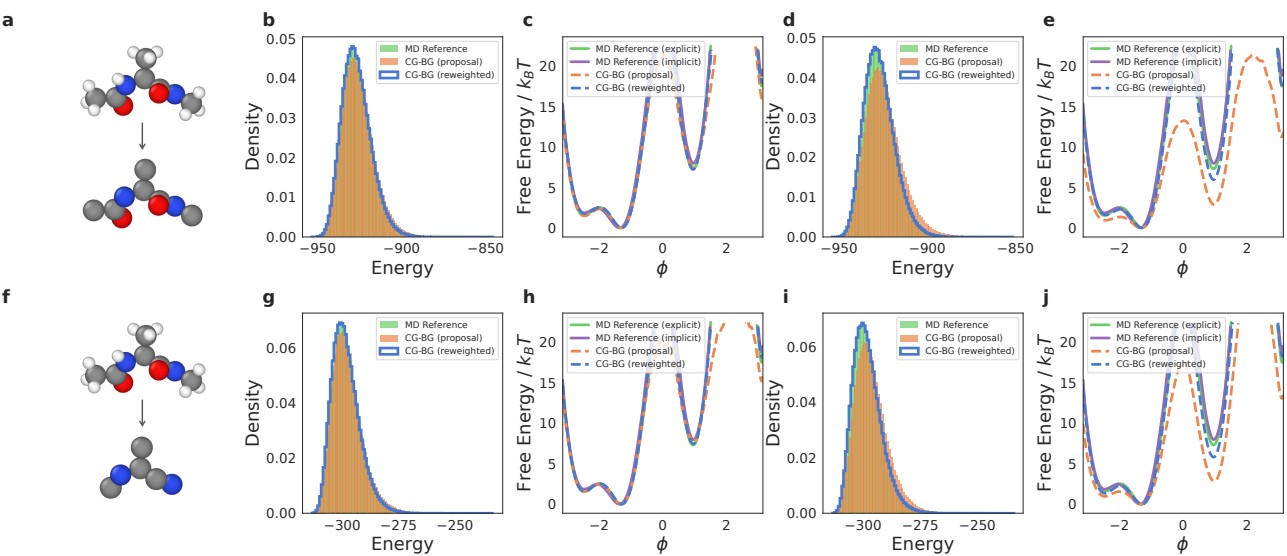

*Figure 2.* **CG-BGs on alanine dipeptide.** (a–e) Heavy Atom mapping results. (a) Heavy Atom mapping. (b) Potential energy distribution under the learned PMF for flow trained on 500 ns *unbiased* data, before and after reweighting, compared with MD reference. (c) $\phi$ dihedral free energy profile for the same model. (d) Energy distribution for flow trained on 10 ns *WT-MetaD* ($\gamma = 1.5$) data. (e) Corresponding $\phi$ dihedral free energy profile. (f–j) Core Beta mapping results are shown in the second row with the same structure as (a–e).

with MD references. The reweighted distributions accurately reproduce relative basin populations and match the reference distribution in transition regions. This is further illustrated by the Ramachandran plots (Fig. 3, Fig. 4e and Fig. 4j). Unreweighted proposals exhibit noisy samples across transition areas and metastable basins (Fig. 8 and Fig. 9), which are assigned low importance weights and effectively filtered out after reweighting. Quantitative metrics in Tab. 2 confirm close agreement between reweighted samples and target equilibrium ensemble.

We further evaluate CG-BGs trained on biased or short non-equilibrium trajectories (umbrella sampling for MB and 10 ns WT-MetaD with $\gamma = 1.5$ for alanine dipeptide), reflecting realistic settings where long unbiased simulations are infeasible. As shown in Fig. 2 and Fig. 6f, while the flow proposals exhibit larger deviations from the MD reference, importance reweighting consistently recovers accurate equilibrium statistics (Tab. 2).

Notably, after reweighting, CG-BGs outperform implicit solvent MD baselines (Tab. 2), highlighting the advantage of learning PMFs from explicit solvent simulations. While implicit solvent models perform reasonably well for alanine dipeptide, we observe increasing differences for tripeptide (Fig. 4c) and hexapeptide (Fig. 4h), consistent with previous observations for more complex molecular systems (Chen et al., 2021). This constitutes an improvement over atomistic BG approaches, which rely on implicit solvent models for reweighting and are therefore fundamentally limited by solvent approximation error. In other words, atomistic BGs can at best achieve the accuracy of implicit solvent baselines (Tab. 2 and Tab. 7). These results show that CG-BGs generate CG equilibrium samples consistent with the target distribution, without requiring long unbiased MD simulations, and offer a practical route to correct systematic biases in existing Boltzmann Emulators.

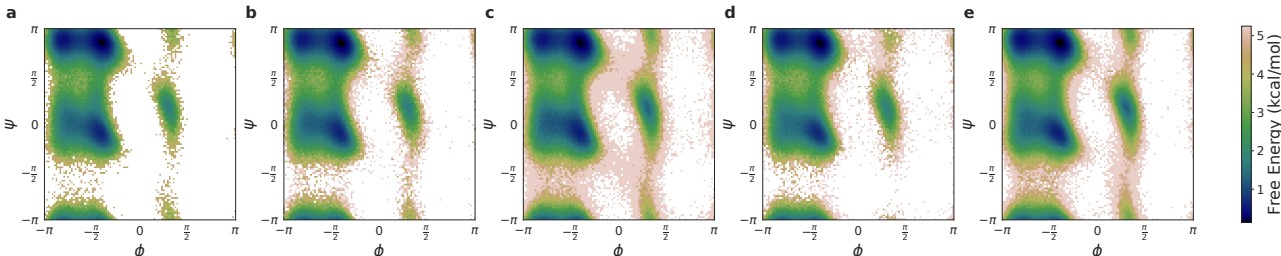

*Figure 3.* **Ramachandran plots of alanine dipeptide.** (a) MD reference. (b,c) Heavy Atom mapping: reweighted Ramachandran distributions. In (b), the flow model is trained on 500 ns *unbiased* data; in (c), it is trained on 10 ns *WT-MetaD* ($\gamma = 1.5$) data. (d,e) Core Beta mapping: same training setups as (b,c). Unreweighted Ramachandran plots are provided in Fig. 8.

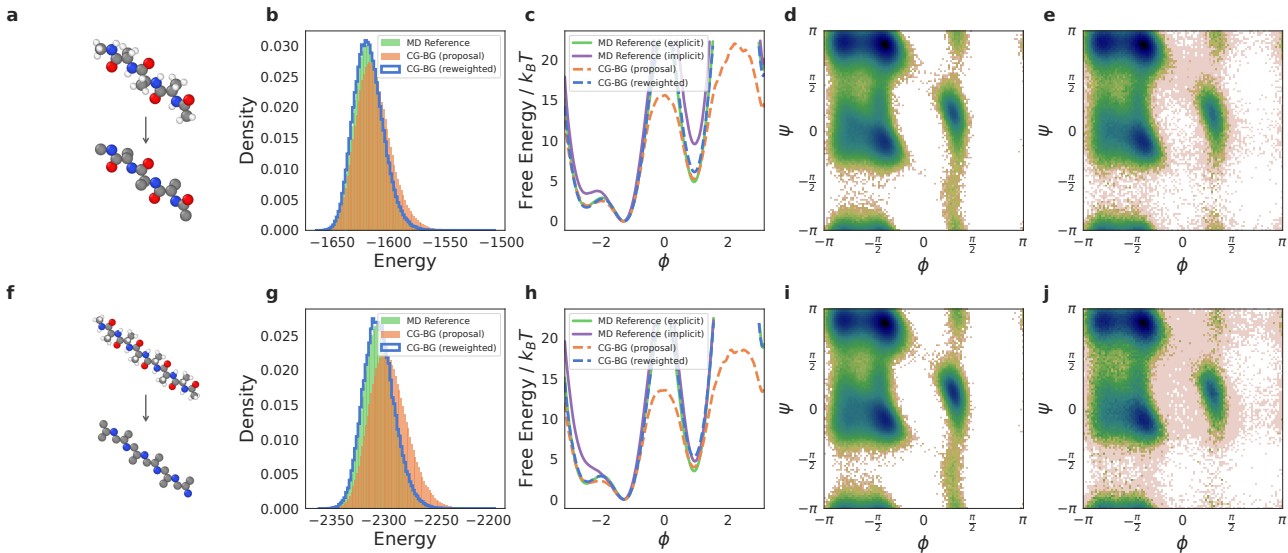

*Figure 4.* **CG-BGs on alanine tripeptide and hexapeptide.** (a–e) Alanine tripeptide (Heavy Atom mapping): (a) CG representation, (b) potential energy distribution, (c) $\phi$ free energy profile, (d) MD reference Ramachandran plot, and (e) reweighted Ramachandran distribution. (f–j) Alanine hexapeptide (Core Beta mapping): (f) CG representation, (g) potential energy distribution, (h) $\phi$ free energy profile, (i) MD reference Ramachandran plot, and (j) reweighted Ramachandran distribution. Unreweighted Ramachandran plots are shown in Fig. 9. The reported free energy profiles and Ramachandran plots correspond to the dihedral pair biased during WT-MetaD: the second pair (of three) for alanine tripeptide and the third pair (of six) for alanine hexapeptide, counting from the N-methyl terminus.

### 4.2. Effect of Coarse-Graining Resolution on Accuracy and Efficiency

We next examine how the choice of CG resolution affects both sampling quality and computational efficiency. To this end, we consider a coarser Core Beta mapping for alanine dipeptide (Fig. 2f) and tripeptide (Fig. 7a). Despite the reduced resolution, CG-BGs trained with the Core Beta mapping remain capable of recovering equilibrium statistics after reweighting (Fig. 2h, 2j and Fig. 7c), with quantitative metrics reported in Tab. 2 and Tab. 3.

Compared to the Heavy Atom mapping, the lower-dimensional Core Beta representation generally yields higher ESS, indicating improved overlap between the flow proposal and the target distribution. This is expected, as generative modeling and importance sampling become easier in lower-dimensional spaces. Nevertheless, after reweighting, the resulting equilibrium statistics are generally less accu-

rate than those obtained with the Heavy Atom mapping. A likely explanation is the increased degeneracy introduced by coarse-graining. For a given CG coordinate $\mathbf{R}$, many atomistic microstates $\mathbf{r}_i$ satisfy $\Xi(\mathbf{r}_i) = \mathbf{R}$ while exerting different projected forces $\mathcal{F}_{\text{proj}}(\mathbf{r}_i)$. As a result, the conditional mean force $\mathbb{E}_{p(\mathbf{r}|\mathbf{R})}[\mathcal{F}_{\text{proj}}(\mathbf{r})]$ exhibits larger variance under coarser mappings, increasing the difficulty of accurately learning the PMF through force matching. This effect is expected to become more pronounced as molecular complexity increases (Görlich & Zavadlav, 2025).

We further benchmark the computational cost of CG-BGs across different resolutions (Tab. 1). As expected, the Core Beta mapping substantially reduces both training and inference time compared to the Heavy Atom and full atomistic representations, with particularly large gains at inference due to cheaper Jacobian evaluations. Note that the reported all atom baseline generates only solute coordinates; generating full configurations with explicit solvent, which would be the proper reference, is computationally infeasible. Overall, the results illustrate a trade-off between statistical efficiency and representation fidelity: coarser mappings improve proposal quality and computational efficiency, while finer mappings show more accurate equilibrium estimates after reweighting. Additional comparisons with prior work are provided in Tab. 5.

*Table 1.* Training and inference time across different CG mappings on alanine dipeptide. Inference times correspond to $10^4$ generated samples. *All Atom* denotes generating the full solute configuration without solvent.

| Stage | **Core Beta** | **Heavy Atom** | **All Atom** |
|---|---|---|---|
| Training | 0.45h | 0.80h | 2.55h |
| Inference | 0.95min | 3.78min | 14.91min |
| Total | 0.47h | 0.86h | 2.80h |

## 4.3. Simulation-Free Evaluation of Learned PMFs

Beyond equilibrium sampling, CG-BGs provide a form of amortized equilibrium benchmarking for learned PMFs. Whereas conventional BGs use a learned proposal distribution to estimate observables under a fixed target energy, the same proposal distribution can also be reused to evaluate and compare multiple candidate PMFs through importance reweighting. Once a sufficiently accurate proposal has been learned, equilibrium observables under different PMFs can be estimated from a single set of generated samples, without performing additional simulations.

Specifically, the flow model generates CG configurations from a proposal distribution approximating the equilibrium ensemble, and importance reweighting maps these samples to the Boltzmann distribution induced by a given PMF. This enables rapid, simulation-free assessment of candidate CG potentials, in contrast to traditional validation pipelines that require separate MD simulations for each model.

We leverage this capability to compare PMFs trained under different data regimes, contrasting models learned from long unbiased MD trajectories ($\mathrm{PMF}_U$) with those trained on a rapidly converged biased dataset ($\mathrm{PMF}_B$). Quantitative metrics from reweighted samples (Tab. 2) provide a direct measure of model accuracy, while dihedral free energy profiles and Ramachandran plots enable visual comparison with atomistic references. Consistent with previous observations (Chen et al., 2026; Görlich & Zavadlav, 2025), $\mathrm{PMF}_U$ (Fig. 5) fails to recover the correct metastable populations along $\phi$, whereas $\mathrm{PMF}_B$ exhibits improved agreement.

Although demonstrated here for CG models, the same principle naturally extends to atomistic machine learning potentials. More generally, generated configurations can be reused to compare multiple learned energy functions without additional simulations, making CG-BGs a practical tool for rapid model validation.

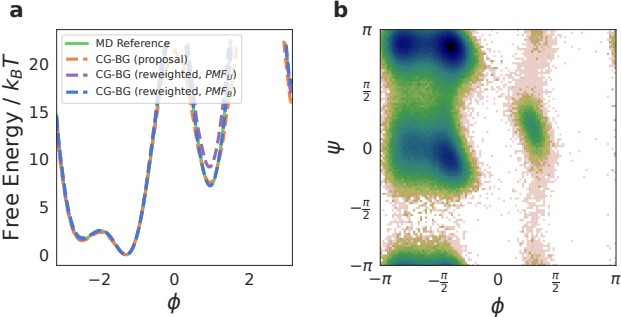

*Figure 5.* **Simulation-free benchmarking of learned CG PMFs using CG-BGs on alanine dipeptide (Heavy Atom).** (a) $\phi$ free energy profile after reweighting with PMFs trained on unbiased ($\mathrm{PMF}_U$) and rapidly converged biased datasets ($\mathrm{PMF}_B$), compared with the MD reference and flow proposal (trained on unbiased data). (b) Ramachandran plot reweighted using $\mathrm{PMF}_U$.

*Table 2.* Quantitative comparison for alanine dipeptide across CG resolutions and baseline models. CG-BGs results are reported after reweighting; *Biased* denotes flow trained on WT-MetaD datasets, while $\mathrm{PMF}_U$ indicates PMFs learned from long unbiased MD data. Flow proposal results are provided in Tab. 7.

| Model | JS ($\downarrow$) | PMF ($\downarrow$) | ESS ($\uparrow$) |
|---|---|---|---|
| | CG-BGs (Reweighted) | | |
| Heavy Atom | **0.0048(1)** | **0.2005(63)** | 0.5112(4) |
| Heavy Atom (Biased) | 0.0063(1) | 0.2277(66) | 0.4115(4) |
| Core Beta | 0.0052(1) | 0.2210(65) | **0.5528(4)** |
| Core Beta (Biased) | 0.0057(1) | 0.2093(58) | 0.4818(4) |
| Heavy Atom ($\mathrm{PMF}_U$) | 0.0050(1) | 0.2131(68) | 0.5196(4) |
| | Implicit Solvent Baselines | | |
| GB (OBC1) | 0.0157(2) | 0.3709(92) | – |
| GB (OBC2) | 0.0182(2) | 0.4028(95) | – |

## 5. Related Work

**Generative models for equilibrium molecular sampling**. Generative models have emerged as powerful tools for sampling Boltzmann distributions (Olsson, 2026; Klein et al., 2023; Rotskoff, 2024; Aranganathan et al., 2025; Janson & Feig, 2025; Xie et al., 2026). Subsequent work has improved BGs in various ways (von Klitzing et al., 2025; Schebek & Rogal, 2025; OuYang et al., 2026), including the incorporation of inductive biases (Köhler et al., 2020), improved transferability across chemical and thermodynamic conditions (Klein & Noé, 2024; Dibak et al., 2022; Moqvist et al., 2025; Invernizzi et al., 2022), and more scalable architectures and likelihood estimators (Tan et al., 2025a; Zhai et al., 2024; Gloy & Olsson, 2025; Rehman et al., 2025; Peng & Gao, 2025). Several approaches address the resolution gap between coarse-grained and atomistic representations through backmapping (Chennakesavalu et al., 2023; Hummerich et al., 2025; Wang et al., 2022) or other reconstruction strategies (Schopmans & Friederich, 2024; Stupp & Koutsourelakis, 2025). Closely related to our setting, Tamagnone et al. (2024) combine a normalizing flow over collective variables with nonequilibrium dynamics to evolve the remaining degrees of freedom. Kohler et al. (2023) instead use a normalizing flow to model the coarse-grained distribution and generate configurations and forces for training coarse-grained MLPs via force matching. Another active line of research considers neural samplers for sampling unnormalized densities (Akhound-Sadegh et al., 2024; Midgley et al., 2023; He et al., 2025; Potaptchik et al., 2025; Liu et al., 2025), which also have applications in molecular systems (Nam et al., 2025; Havens et al., 2025; Blessing et al., 2026).

**Coarse-grained machine learning potentials**. The development of CG MLPs can be viewed as an extension of

*Table 3.* Quantitative comparison for alanine tripeptide and hexapeptide across CG resolutions and baseline models. CG-BGs results are reported after reweighting. Flow proposal results are provided in Tab. 8 and Tab. 9.

| Model | Alanine Tripeptide | | | Alanine Hexapeptide | | |
|---|---|---|---|---|---|---|
| | **JS** ($\downarrow$) | **PMF** ($\downarrow$) | **ESS** ($\uparrow$) | **JS** ($\downarrow$) | **PMF** ($\downarrow$) | **ESS** ($\uparrow$) |
| Core Beta | 0.0060(1) | 0.2112(51) | **0.4212(5)** | 0.0100(1) | 0.3646(81) | 0.1231(3) |
| Heavy Atom | **0.0056(1)** | **0.1957(52)** | 0.3201(4) | — | — | — |
| Implicit Solvent Baselines | | | | | | |
| GB (OBC2) | 0.0932(3) | 1.0274(65) | — | 0.1652(3) | 1.8401(70) | — |

broader effort to construct accurate machine learning interatomic potentials from first-principles calculations. (Batzner et al., 2022; Batatia et al., 2022; Unke et al., 2021). Beyond *bottom-up* approaches (Jin et al., 2022; Noid, 2013), CG potentials can also be parameterized using *top-down* methods that reproduce macroscopic observables or experimental measurements (Marrink et al., 2007; Thaler & Zavadlav, 2021; Fuchs & Zavadlav, 2025). Recent work has explored different machine learning approaches for learning CG MLPs (Zhang et al., 2018; Wang et al., 2019; Kohler et al., 2023; Arts et al., 2023; Plainer et al., 2025). Despite these advances, learning transferable and computationally efficient CG MLPs remains challenging (Charron et al., 2025; Mirarchi et al., 2024; Majewski et al., 2023; Durumeric et al., 2023). A limitation of standard force matching objectives is their strong reliance on large amounts of converged simulation data. ESFM (Chen et al., 2026) addresses this by learning from rapidly converged biased simulations. Alternative strategies include constrained MD approaches for approximating mean forces prior to training (Ciccotti et al., 2005; Park et al., 2026; Fan et al., 2026).

## 6. Conclusion

This work introduces CG-BGs, a scalable framework for equilibrium sampling of coarse-grained molecular systems. By targeting the marginal equilibrium distribution defined by the PMF, CG-BGs reduce the effective dimensionality of the sampling problem while retaining asymptotic correctness through importance reweighting. The underlying PMF can be learned from rapidly converged data using enhanced sampling force matching, providing a correction mechanism for existing CG Boltzmann Emulators. Even at high levels of coarse-graining, CG-BGs capture solvent-mediated and many-body effects, and enable one-shot, simulation-free evaluation of CG MLPs.

**Limitations**. The current approach uses predefined collective variables for coarse-graining and enhanced sampling, which may be nontrivial to identify for complex systems. Recent advances in collective variable discovery (Zhang et al., 2024; Ribeiro et al., 2018; Chen & Ferguson, 2018;

Herringer et al., 2023; Mehdi et al., 2024) and uncertainty quantification (Zaverkin et al., 2024; Musil et al., 2019) provide promising avenues to address these challenges.

**Future work**. Extending CG-BGs to larger, more complex systems is a natural next step, leveraging demonstrated transferability of generative models (Didi et al., 2026; Antoniadis et al., 2026) and MLPs (Wood et al., 2025; Kabylda et al., 2025). More broadly, advances in exact-likelihood generative modeling, including autoregressive architectures (Rehman et al., 2026; Yu et al., 2026), as well as improvements in atomistic Boltzmann Generators, can be readily transferred to the CG setting. The simulation-free evaluation method introduced here also opens the possibility of elucidating the design space of coarse-grained machine learning potentials. It could enable systematic assessment of how architectural choices, parameterizations, and training objectives influence PMF accuracy. In contrast to conventional pipelines that rely on validation loss or repeated MD simulations, this approach could provide direct assessment of model quality via observable level comparisons. Finally, while CG-BGs are trained from simulation data in this work, one could also explore energy-based training or more general neural sampler formulations using the learned PMFs as unnormalized targets.

## Acknowledgements

We thank Simon Olsson, Franz Görlich, Nuno Costa, and Paul Fuchs for fruitful discussions and helpful feedback. This work was funded by the European Union through the ERC (StG SupraModel) - 101077842. Views and opinions expressed are however those of the author(s) only and do not necessarily reflect those of the European Union or the European Research Council Executive Agency. Neither the European Union nor the granting authority can be held responsible for them.

## Impact Statement

This work focuses on sampling from Boltzmann distributions, a problem of broad interest in machine learning and

AI for Science, with applications in both statistical physics and molecular simulations. We introduce Coarse-Grained Boltzmann Generators, which can be trained on molecular systems and applied to tasks such as drug and material discovery. While we do not anticipate immediate negative impacts, we encourage careful consideration when scaling these methods to prevent potential misuse.

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

# A. Proofs

## A.1. Proof of Proposition 1

**Proposition 1.** *Let $p^*(\mathbf{R}) \propto e^{-\beta U^*(\mathbf{R})}$ be the true marginal and $p_\eta(\mathbf{R}) \propto e^{-\beta U_\eta(\mathbf{R})}$ the learned distribution. If $p^*$ satisfies a Logarithmic Sobolev Inequality (LSI) with constant $\rho > 0$. Then, the Kullback-Leibler divergence between the learned and true distributions is bounded by the expected squared force error:*

$$\mathcal{D}_{\mathrm{KL}}(p_\eta \| p^*) \leq \frac{\beta^2}{2\rho} \mathbb{E}_{p_\eta} \left[ \|\nabla U_\eta(\mathbf{R}) - \nabla U^*(\mathbf{R})\|^2 \right]. \tag{14}$$

*Proof.* The Kullback-Leibler divergence is defined as:

$$\mathcal{D}_{\mathrm{KL}}(p_\eta \| p^*) = \int p_\eta(\mathbf{R}) \log \frac{p_\eta(\mathbf{R})}{p^*(\mathbf{R})} d\mathbf{R}. \tag{19}$$

The Fisher Divergence (or relative Fisher information) between $p_\eta$ and $p^*$ is defined as:

$$\mathcal{J}(p_\eta \| p^*) = \int p_\eta(\mathbf{R}) \left\| \nabla \log p_\eta(\mathbf{R}) - \nabla \log p^*(\mathbf{R}) \right\|^2 d\mathbf{R}. \tag{20}$$

Since $p_\eta(\mathbf{R}) = Z_\eta^{-1} e^{-\beta U_\eta(\mathbf{R})}$ and $p^*(\mathbf{R}) = (Z^*)^{-1} e^{-\beta U^*(\mathbf{R})}$, the gradients of the log-densities are directly proportional to the forces:

$$\nabla \log p_\eta(\mathbf{R}) = -\beta \nabla U_\eta(\mathbf{R}), \quad \nabla \log p^*(\mathbf{R}) = -\beta \nabla U^*(\mathbf{R}). \tag{21}$$

Substituting these into the definition of the Fisher Divergence:

$$\mathcal{J}(p_\eta \| p^*) = \int p_\eta(\mathbf{R}) \left\| (-\beta \nabla U_\eta(\mathbf{R})) - (-\beta \nabla U^*(\mathbf{R})) \right\|^2 d\mathbf{R} \tag{22}$$

$$= \beta^2 \int p_\eta(\mathbf{R}) \left\| \nabla U_\eta(\mathbf{R}) - \nabla U^*(\mathbf{R}) \right\|^2 d\mathbf{R} \tag{23}$$

$$= \beta^2 \mathbb{E}_{p_\eta} \left[ \|\mathcal{F}_\eta(\mathbf{R}) - \mathcal{F}^*(\mathbf{R})\|^2 \right]. \tag{24}$$

We assume that the target distribution $p^*$ satisfies a Logarithmic Sobolev Inequality (LSI) with constant $\rho > 0$. By definition, this inequality implies that for any distribution $p_\eta$ absolutely continuous with respect to $p^*$:

$$\mathcal{D}_{\mathrm{KL}}(p_\eta \| p^*) \leq \frac{1}{2\rho} \mathcal{J}(p_\eta \| p^*). \tag{25}$$

Substituting our expression for the Fisher Divergence into the LSI yields the final bound:

$$\mathcal{D}_{\mathrm{KL}}(p_\eta \| p^*) \leq \frac{1}{2\rho} \left( \beta^2 \mathbb{E}_{p_\eta} \left[ \|\nabla U_\eta(\mathbf{R}) - \nabla U^*(\mathbf{R})\|^2 \right] \right). \tag{26}$$

Dividing out the constants concludes the proof. $\square$

## A.2. Proof of Proposition 2

**Proposition 2.** (Chen et al. (2026)) *Let $V(\mathbf{R})$ be a bias potential depending only on the coarse-grained coordinates. The conditional distribution of atomistic configurations given $\mathbf{R}$ is invariant:*

$$p_V(\mathbf{r} \mid \mathbf{R}) = p(\mathbf{r} \mid \mathbf{R}). \tag{15}$$

*Proof.* Let $p(\mathbf{r}) = Z^{-1} e^{-\beta u(\mathbf{r})}$ be the unbiased equilibrium distribution. By definition, the unbiased marginal distribution is

$$p(\mathbf{R}) = \int p(\mathbf{r}) \delta(\Xi(\mathbf{r}) - \mathbf{R}) d\mathbf{r} = \frac{1}{Z} \int e^{-\beta u(\mathbf{r})} \delta(\Xi(\mathbf{r}) - \mathbf{R}) d\mathbf{r}. \tag{27}$$

Rearranging this yields the identity for the unnormalized marginal:

$$\int e^{-\beta u(\mathbf{r})} \delta(\Xi(\mathbf{r}) - \mathbf{R}) d\mathbf{r} = Z p(\mathbf{R}). \tag{28}$$

Now, consider the biased distribution $p_V(\mathbf{r}) = Z_V^{-1} e^{-\beta(u(\mathbf{r}) + V(\Xi(\mathbf{r})))}$. The biased marginal distribution is:

$$p_V(\mathbf{R}) = \int \frac{e^{-\beta u(\mathbf{r})} e^{-\beta V(\Xi(\mathbf{r}))}}{Z_V} \delta(\Xi(\mathbf{r}) - \mathbf{R}) d\mathbf{r} \tag{29}$$

$$= \frac{e^{-\beta V(\mathbf{R})}}{Z_V} \underbrace{\int e^{-\beta u(\mathbf{r})} \delta(\Xi(\mathbf{r}) - \mathbf{R}) d\mathbf{r}}_{=Z p(\mathbf{R}) \text{ (from Eq. 28)}} \tag{30}$$

$$= \frac{Z}{Z_V} e^{-\beta V(\mathbf{R})} p(\mathbf{R}). \tag{31}$$

Finally, substituting this into the definition of the conditional distribution:

$$p_V(\mathbf{r} \mid \mathbf{R}) = \frac{p_V(\mathbf{r}) \delta(\Xi(\mathbf{r}) - \mathbf{R})}{p_V(\mathbf{R})} = \frac{Z_V^{-1} e^{-\beta u(\mathbf{r})} e^{-\beta V(\mathbf{R})} \delta(\dots)}{\frac{Z}{Z_V} e^{-\beta V(\mathbf{R})} p(\mathbf{R})} = \frac{e^{-\beta u(\mathbf{r})} \delta(\dots)}{Z p(\mathbf{R})} = p(\mathbf{r} \mid \mathbf{R}). \tag{32}$$

$\square$

## B. Datasets

### B.1. Müller-Brown Potential

**Potential Parameters**. For the two-dimensional toy system, we use a Müller–Brown potential defined as

$$u(x,y) = u_1(x,y) + u_2(x,y) + u_3(x,y) + u_4(x,y), \tag{33}$$

with (Raja et al., 2025)

$$u_1(x,y) = -17.3 \exp\left[-0.0039(x-48)^2 - 0.0391(y-8)^2\right],$$
$$u_2(x,y) = -8.7 \exp\left[-0.0039(x-32)^2 - 0.0391(y-16)^2\right],$$
$$u_3(x,y) = -14.7 \exp\left[-0.0254(x-24)^2 + 0.043(x-24)(y-32) - 0.0254(y-32)^2\right],$$
$$u_4(x,y) = 1.3 \exp\left[0.00273(x-16)^2 + 0.0023(x-16)(y-24) + 0.00273(y-24)^2\right].$$

**Umbrella Sampling**. For umbrella sampling, we introduce a biasing potential along the $x$ coordinate,

$$V_x(x) = -4 \exp\left[-\frac{(x-32.0)^2}{2 \cdot 5^2}\right].$$

This enables better sampling of the configurations around $\mathbf{x}_0 = 32$, allowing a better representation of transition regions that are otherwise rarely visited in unbiased trajectories.

**Simulation Details**. For the MB dataset generation, we perform two-dimensional Langevin dynamics with a time step of 0.1, mass $m = 1.0$, friction coefficient $\gamma = 0.1$, and temperature $k_\mathrm{B} T = 1.0$. Ten independent trajectories of length $10^7$ steps are generated, with initial positions sampled uniformly from $[10, 50]^2$ and initial velocities drawn from a Gaussian distribution with standard deviation 0.1. Configurations are recorded every 10 steps. The dynamics follow

$$m\ddot{\mathbf{r}} = -\nabla(u(\mathbf{r}) + V(x)) - \gamma m \dot{\mathbf{r}} + \sqrt{2\gamma k_\mathrm{B} T m}\, \boldsymbol{\eta}(t), \tag{34}$$

where $V(x)$ is the applied bias, and $\boldsymbol{\eta}(t)$ denotes Gaussian white noise. For each saved configuration, unbiased forces from $\nabla u$ are computed and stored.

## B.2. Alanine Peptides

**Force Fields**. For the explicit solvent dataset, alanine peptide systems are parameterized using the AMBER99SB-ILDN force field (Lindorff-Larsen et al., 2010) and solvated in a cubic box of TIP3P water molecules. Explicit solvent simulations of alanine dipeptide are carried out with `GROMACS` (Van Der Spoel et al., 2005), while all remaining peptide systems are simulated using `OpenMM` (Eastman et al., 2023). For implicit solvent, the same force field is used together with the generalized Born (OBC1/OBC2) model, and all simulations are performed using `OpenMM`.

**Well-Tempered Metadynamics**. We perform well-tempered metadynamics (WT-MetaD) (Barducci et al., 2008) simulations of alanine peptides in explicit solvent using `GROMACS` coupled with `PLUMED` (Bonomi et al., 2009). The backbone dihedral angles $\phi$ (C–N–C$_\alpha$–C) and $\psi$ (N–C$_\alpha$–C–N) are chosen as collective variables. Gaussian hills with height $1.2$ kJ/mol and width $0.35$ rad are deposited every 500 integration steps. Datasets are generated with bias factors $\gamma = 1.5$ and $\gamma = 9$, where $\gamma = 1.5$ is used to train flow model for biased dipeptide proposal generation and $\gamma = 9$ is used for enhanced sampling force matching. Positions and forces are recorded. To ensure unbiased force labels, all forces are recomputed by rerunning the saved trajectories in `GROMACS` without the metadynamics bias using the `mdrun -rerun` functionality. This guarantees that each configuration is associated with forces from the underlying unbiased potential.

For WT-MetaD, the backbone dihedral pair $(\phi, \psi)$ is used as collective variables for the dipeptide system. For the tripeptide and hexapeptide systems, biasing is applied only to the specific dihedral pair of interest, which is the second pair (of three) for alanine tripeptide and the third pair (of six) for alanine hexapeptide, counting from the N-methyl terminus. As a result, convergence is primarily enforced along the targeted collective variables, which we find sufficient for accurate estimation of the PMF along the corresponding degrees of freedom in this work. More general biasing schemes or longer simulations may further improve sampling of the remaining degrees of freedom, and thereby improve the accuracy of the global PMF.

**Simulation Details**. All simulations are performed in the NVT ensemble at a temperature of 300 K, with a time step of $0.5$ fs and no bond constraints. After energy minimization, production dynamics are carried out using a velocity-rescale thermostat (time constant 0.1 ps). Long-range electrostatics are treated using the particle mesh Ewald method, and van der Waals interactions are truncated at 1.0 nm. We summarize the dataset configurations used in this work in Tab. 4.

*Table 4.* Overview of simulation details for alanine peptide datasets.

| System | Solvent | Dataset | Method | Length |
|--------|---------|---------|--------|--------|
| Dipeptide | Explicit (TIP3P) | Unbiased MD | None | 500 ns |
| | Explicit (TIP3P) | Biased MD (CNF) | WT-MetaD ($\gamma = 1.5$) | 10 ns |
| | Explicit (TIP3P) | Biased MD (PMF) | WT-MetaD ($\gamma = 9$) | 10 ns |
| | Implicit (OBC1) | Unbiased MD | None | 500 ns |
| | Implicit (OBC2) | Unbiased MD | None | 500 ns |
| Tripeptide | Explicit (TIP3P) | Unbiased MD | None | 1000 ns |
| | Explicit (TIP3P) | Biased MD (PMF) | WT-MetaD ($\gamma = 9$) | 50 ns |
| | Implicit (OBC2) | Unbiased MD | None | 1500 ns |
| Hexapeptide | Explicit (TIP3P) | Unbiased MD | None | 1500 ns |
| | Explicit (TIP3P) | Biased MD (PMF) | WT-MetaD ($\gamma = 9$) | 100 ns |
| | Implicit (OBC2) | Unbiased MD | None | 1500 ns |

# C. Experimental Details for Conditional Normalizing Flows

## C.1. Architecture

**Müller-Brown Potential**. For MB potential, we use a multilayer perceptron augmented with time conditioning for flow matching. The network consists of three hidden layers with width 96. Flow time is embedded into a 16-dimensional time embedding and concatenated with the input.

**Alanine Peptides**. For peptides, we use an adapted Graph Transformer architecture (Plainer et al., 2025; Arts et al., 2023;

Shi et al., 2020). Given bead positions $\mathbf{x}_i$ and bead features $\mathbf{h}_i$, edge attributes are constructed as

$$\mathbf{d}_{ij} = \mathbf{x}_i - \mathbf{x}_j, \quad r_{ij} = \|\mathbf{d}_{ij}\|, \quad \mathbf{e}_{ij} = [\mathbf{d}_{ij}, r_{ij}], \tag{35}$$

and node attributes are initialized as

$$\mathbf{n}_i^{(0)} = [\mathbf{h}_i, \mathbf{x}_i, t], \tag{36}$$

where $t$ denotes the flow time. The bead features and time are embedded into 16- and 4-dimensional vectors, respectively, concatenated with positions, and projected to 128-dimensional node embeddings via a linear layer. Edge attributes are also embedded into 128 dimensions. The model consists of three Graph Transformer layers with 8 attention heads and a head dimension of 64. To enforce rotational equivariance, random global rotations are applied to molecular configurations during training as data augmentation (Abramson et al., 2024). Additionally, translational equivariance is guaranteed by moving the center of mass to the origin and adding noise to the center of mass to lift the data dimensionality back. (Tan et al., 2025a).

### C.2. Training Configuration

All models are trained using AdamW with weight decay $10^{-5}$ and a cosine learning-rate schedule decreasing from $3 \times 10^{-4}$ to $1 \times 10^{-5}$. For the Müller–Brown potential, training is performed for 1000 epochs with batch size 256 using 20k training samples. For alanine dipeptide, models are trained for 5000 epochs with batch size 1024 using 50k samples. For alanine tripeptide and hexapeptide, models are trained for 10000 epochs with batch size 1024 using 200k samples. We find no consistent benefit from exponential moving average (EMA) of model parameters and therefore do not employ it in our experiments.

### C.3. Inference

As Hutchinson's trace estimator introduces bias for BGs (Tan et al., 2025a; Peng & Gao, 2025), we compute the divergence exactly using automatic differentiation. For inference, we use the Dormand–Prince 5(4) method (`dopri5`) with absolute and relative tolerances set to $10^{-5}$. Inference is performed with a batch size of 500.

### C.4. Computational Cost

The reported inference times (Tab. 5) correspond to generating $10^4$ samples. All other training and inference parameters are provided in §C.2 and §C.3.

*Table 5.* Training and inference time for CFM across different systems and settings. Inference times correspond to $10^4$ generated samples. Results marked with $^\dagger$ are taken from (Rehman et al., 2025) and are included for reference. Reported runtimes should be interpreted in light of differences in hardware, implementation details, and training and inference batch sizes. Note also that peptide nomenclature differs between the two works due to different conventions for counting terminal capping groups: our alanine tripeptide and hexapeptide correspond to the alanine tetrapeptide and heptapeptide systems reported in (Rehman et al., 2025), respectively.

| Model | Alanine Dipeptide | | | Alanine Tripeptide | | |
|---|---|---|---|---|---|---|
| | **Training** | **Inference** | **Total** | **Training** | **Inference** | **Total** |
| Core Beta | 0.45h | 0.95min | 0.47h | 7.32h | 6.47min | 7.43h |
| Heavy Atom | 0.80h | 3.78min | 0.86h | 17.63h | 14.22min | 17.87h |
| All Atom | 2.55h | 14.91min | 2.80h | — | — | — |
| ECNF++$^\dagger$ | — | — | 12.52h | — | — | 32.17h |
| SBG$^\dagger$ | — | — | 16.83h | — | — | 41.67h |
| DiT CNF$^\dagger$ | — | — | 9.56h | — | — | 24.10h |
| FALCON$^\dagger$ | — | — | 7.65h | — | — | 25.76h |

| Model | Alanine Hexapeptide | | |
|---|---|---|---|
| | **Training** | **Inference** | **Total** |
| Core Beta | 23.01h | 20.07min | 23.34h |

# D. Experimental Details for Force Matching

## D.1. Architecture

**Müller-Brown Potential**. For the MB potential, we use a radial basis function (RBF) feature map followed by a multilayer perceptron. The RBF expansion has $K = 100$ centers, initialized uniformly in $[10, 50]^2$ and optimized during training, with a fixed width $\sigma = 5.0$. The features are passed through four fully connected layers of size 128 with softplus activation, followed by a linear output layer.

**Alanine Peptides**. For peptides, the CG potential $U_\eta(\mathbf{R})$ is parameterized using the MACE architecture (Batatia et al., 2022), an equivariant message-passing graph neural network. Each CG bead is represented as a node in a geometric graph, with edges connecting neighbors within a cutoff radius and encoding relative position vectors. The model uses hidden irreducible representations of $32 \times 0e + 32 \times 1o$, processed through two interaction layers with correlation order 3 and an angular momentum expansion truncated at $\ell_{\max} = 3$. Node features are decoded by a readout layer ($16 \times 0e$) into a scalar energy prediction ($1 \times 0e$). Periodic displacement functions are applied during graph construction to handle boundary conditions correctly.

## D.2. Training Configuration

For MB, training uses the Adam optimizer (via Optax) with a constant learning rate of $10^{-4}$ and batch size 128 for 500 epochs. For peptides, training uses Adam with exponential learning rate decay ($\eta_0 = 10^{-3}$, decay rate 0.01), batch size 256, and 300 epochs. We use 500k training samples for alanine dipeptide and 1000k training samples for alanine tripeptide and hexapeptide across all CG mappings. Training and validation splits are with a 90/10 ratio. Gradients are clipped to a global norm of 1.0. Validation losses in CG force matching are often noisy and do not consistently correlate with potential quality. We use the final training checkpoint for inference in all experiments.

## D.3. Computational Cost

The reported inference times (Tab. 6) correspond to evaluating $10^4$ samples. Training parameters follow §D.2. A batch size of 500 is used for inference.

*Table 6.* Training and inference time for MACE model on alanine dipeptide. Inference times correspond to $10^4$ samples evaluated.

|  | Core Beta | Heavy Atom |
|---|---|---|
| Training | 1.88h | 4.11h |
| Inference | 0.66s | 0.72s |

# E. Compute Infrastructure and Software

## E.1. Hardware

All experiments, including model training, inference, and computational benchmarks, are performed on a single NVIDIA A100 GPU with 80 GB memory.

## E.2. Software

CFM is implemented using JAX, Diffrax (Kidger, 2021), and Flax (Heek et al., 2024). Graph transformer is adapted from the implementation provided in https://github.com/noegroup/ScoreMD/blob/main/src/scoremd/models/graph_transformer.py. Training of CG PMF is carried out using chemtrain (Fuchs et al., 2025b) and chemtrain-deploy (Fuchs et al., 2025a) built on JAX, M.D. (Schoenholz & Cubuk, 2020). Molecular structures are visualized using OVITO (Stukowski, 2009).

# F. Algorithms

---

**Algorithm 1** Training CG PMF via ESFM

---

**Input:** Dataset $\mathcal{D}_{\mathrm{bias}} = \{(\mathbf{r}, \mathcal{F}_{\mathrm{proj}}(\mathbf{r}))\}$ (Rapidly converged); batch size $B$
**Initialize:** CG PMF network $U_\eta$
**while** not converged **do**
    Sample $\{(\mathbf{r}^{(i)}, \mathcal{F}_{\mathrm{proj}}^{(i)})\}_{i=1}^{B} \sim \mathcal{D}_{\mathrm{bias}}$
    Compute $\mathbf{R}^{(i)} \leftarrow \Xi(\mathbf{r}^{(i)})$
    $\mathcal{L}_{\mathrm{ESFM}} \leftarrow \frac{1}{B}\sum_{i=1}^{B} \left\|\nabla_\mathbf{R} U_\eta(\mathbf{R}^{(i)}) - \mathcal{F}_{\mathrm{proj}}^{(i)}\right\|_2^2$
    Update $\eta \leftarrow \mathrm{Optim}\big(\eta, \nabla_\eta \mathcal{L}_{\mathrm{ESFM}}\big)$
**end while**
**Return** $U_\eta$

---

**Algorithm 2** Training Flow Model via CFM

---

**Input:** Dataset $\mathcal{D} = \{\mathbf{r}\}$ (biased or unbiased); batch size $B$
**Initialize:** flow model $v_\theta$
**while** not converged **do**
    Sample $\mathbf{r}^{(i)} \sim \mathcal{D}$
    Compute $\mathbf{R}_1^{(i)} \leftarrow \Xi(\mathbf{r}^{(i)})$
    Sample $\mathbf{R}_0^{(i)} \sim \mathcal{N}(\mathbf{0}, \mathbf{I})$
    Sample $t^{(i)} \sim \mathcal{U}[0, 1]$
    $\mathbf{R}_t^{(i)} \leftarrow (1 - t^{(i)})\mathbf{R}_0^{(i)} + t^{(i)}\mathbf{R}_1^{(i)}$
    $u_t^{(i)} \leftarrow \mathbf{R}_1^{(i)} - \mathbf{R}_0^{(i)}$
    $\mathcal{L}_{\mathrm{CFM}} \leftarrow \frac{1}{B}\sum_{i=1}^{B} \left\|v_\theta(t^{(i)}, \mathbf{R}_t^{(i)}) - u_t^{(i)}\right\|_2^2$
    Update $\theta \leftarrow \mathrm{Optim}\big(\theta, \nabla_\theta \mathcal{L}_{\mathrm{CFM}}\big)$
**end while**
**Return** $v_\theta$

---

**Algorithm 3** Sampling & Reweighting

---

**Input:** Trained models $U_\eta$, $v_\theta$; number of samples $N$
Initialize sample set $\mathcal{X} \leftarrow \emptyset$, log-weights $\mathcal{W} \leftarrow \emptyset$
**for** $i = 1$ to $N$ **do**
    Sample $\mathbf{z} \sim \mathcal{N}(\mathbf{0}, \mathbf{I})$
    Solve ODE $\frac{d\mathbf{R}}{dt} = v_\theta(t, \mathbf{R})$ with $\mathbf{R}(0) = \mathbf{z}$
    $\mathbf{R}^{(i)} \leftarrow \mathbf{R}(1)$
    $\Delta\ell^{(i)} \leftarrow \int_0^1 \nabla \cdot v_\theta(t, \mathbf{R}(t))\, dt$
    $\log q_\theta(\mathbf{R}^{(i)}) \leftarrow \log p_0(\mathbf{z}) - \Delta\ell^{(i)}$
    $E^{(i)} \leftarrow U_\eta(\mathbf{R}^{(i)})$
    $\log \tilde{w}^{(i)} \leftarrow -\beta E^{(i)} - \log q_\theta(\mathbf{R}^{(i)})$
    Append $\mathbf{R}^{(i)}$ to $\mathcal{X}$ and $\log \tilde{w}^{(i)}$ to $\mathcal{W}$
**end for**
Weight clipping for $\mathcal{W}$
Normalize weights $\{w^{(i)}\}$ and compute ESS
**Return** samples $\mathcal{X}$, weights $\{w^{(i)}\}$, ESS

---

# G. Additional Results

## G.1. Müller-Brown Potential

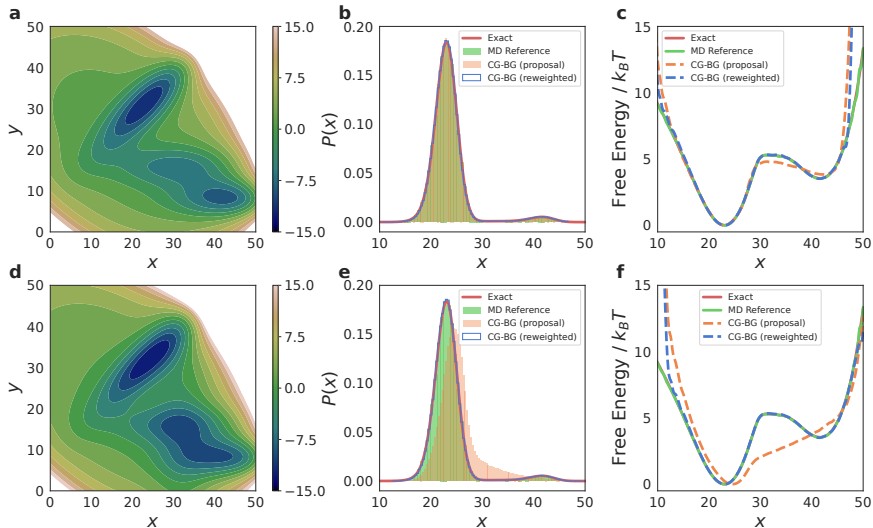

*Figure 6.* **CG-BGs on the MB potential.** (a) Two-dimensional unbiased MB potential energy surface (functional form in §B). (b) Marginal probability density along the $x$ coordinate. (c) Free energy profiles before and after reweighting for CG-BGs, where flow is trained on *unbiased* data, compared with the exact solution and MD reference. (d-f) Same as (a-c), but for flow trained on *biased* data.

## G.2. Alanine Tripeptide (Core Beta mapping)

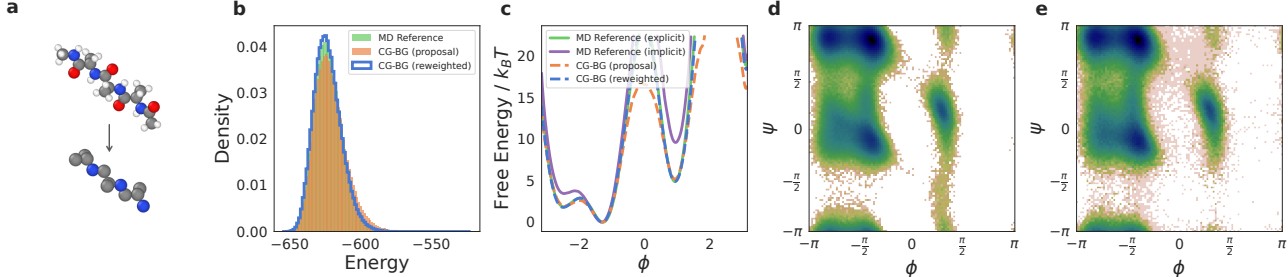

*Figure 7.* **CG-BGs on alanine tripeptide (Core Beta).** (a) Core Beta mapping. (b) Potential energy distribution under the learned PMF. (c) $\phi$ dihedral free energy profile. (d) MD reference Ramachandran plot. (e) Reweighted Ramachandran distribution obtained from CG-BGs.

## G.3. Ramachandran Plots

We show Ramachandran plots for peptides to illustrate the effect of importance reweighting (Fig. 8 and Fig. 9), using MD reference distributions from explicit and implicit solvent simulations. The raw flow proposals exhibit noisy sampling and place probability mass in low-probability regions of the free energy landscape. These configurations receive low importance weights and therefore contribute negligibly after reweighting, yielding distributions that closely match the explicit solvent reference MD ensembles.

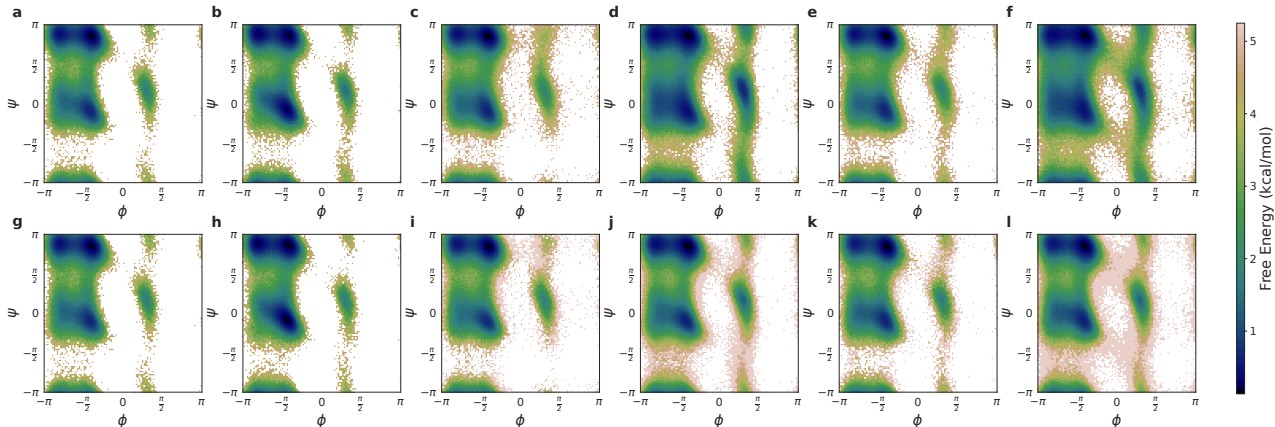

*Figure 8.* **Ramachandran plots for alanine dipeptide.** (a,g) Reference MD (explicit solvent). (b) Implicit solvent MD (OBC1). (h) Implicit solvent MD (OBC2). (c) Core Beta (unbiased) proposal. (d) Core Beta (WT-MetaD) proposal. (e) Heavy Atom (unbiased) proposal. (f) Heavy Atom (WT-MetaD) proposal. (i–l) Reweighted distributions corresponding to (c–f).

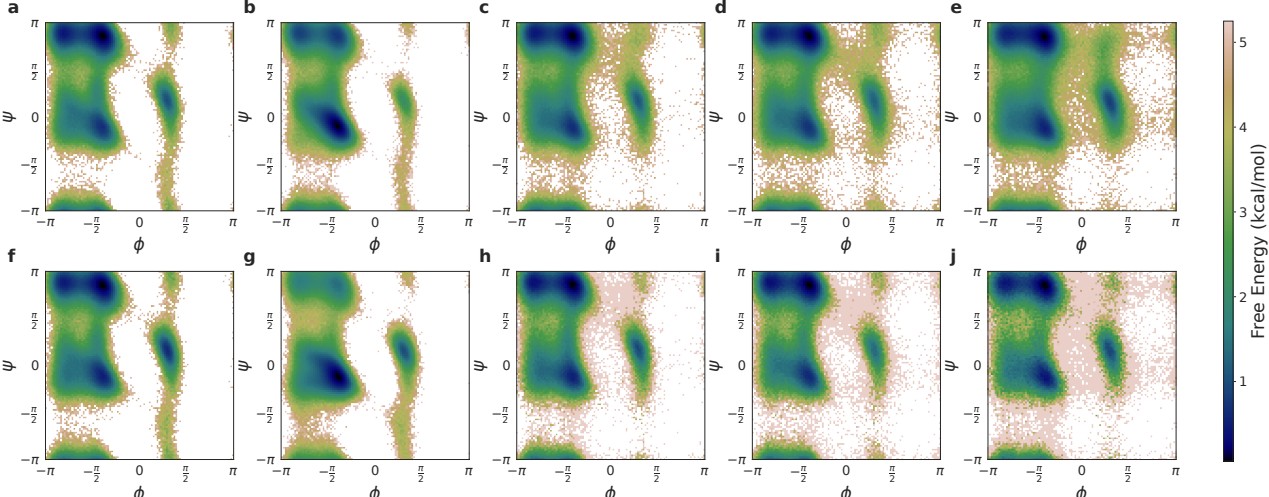

*Figure 9.* **Ramachandran plots for alanine tripeptides and hexapeptides.** (a,b) Reference MD for tripeptide (explicit and implicit solvent). (c,d) Flow proposals for tripeptide (Core Beta and Heavy Atom). (e) Flow proposal for hexapeptide (Core Beta). (f,g) Reference MD for hexapeptide (explicit and implicit solvent). (h–j) Reweighted distributions corresponding to (c–e).

### G.4. Ablation of Weight Clipping

To stabilize importance reweighting, we apply a weight clipping strategy, discarding the top $1\%$ of samples with the largest log-weights.

As shown in Tab. 7, Tab. 8 and Tab. 9, reweighting without clipping leads to large JS divergences and PMF errors, together with ESS, indicating severe weight degeneracy. In contrast, weight clipping restores stable estimates and substantially improves all metrics.

We further analyze sensitivity to the clipping ratio in Fig. 10. Aggressive clipping (10–20%) maximizes ESS but introduces bias in the reweighted distributions. Small clipping ratios (close to 0%) preserve unbiasedness but suffer from high weights variance. Balancing this bias–variance trade-off, we choose a conservative $1\%$ clipping threshold across all experiments, which yields stable metrics while maintaining physically consistent distributions.

*Table 7.* Quantitative comparison for alanine dipeptide across CG resolutions and different weight clipping thresholds. Atomistic BG results from previous studies (Tan et al., 2025a) are included for reference, where models are trained on implicit solvent datasets and evaluated after reweighting using the implicit solvent energy function. Reported values are computed against the explicit solvent MD reference. For alanine dipeptide, TarFlow is trained on biased simulation data, which leads to low ESS.

| Model | JS (↓) | PMF (↓) | ESS (↑) |
|---|---|---|---|
| Flow Trained on *Unbiased* Dataset | | | |
| Heavy Atom Proposal | **0.0048(1)** | 0.2749(72) | — |
| Heavy Atom Reweighted Proposal (1% clip) | **0.0048(1)** | **0.2005(63)** | 0.5112(4) |
| Heavy Atom Reweighted Proposal (0% clip) | 0.0065(1) | 0.2548(91) | 0.4038(28) |
| Core Beta Proposal | 0.0058(1) | 0.3662(85) | — |
| Core Beta Reweighted Proposal (1% clip) | 0.0052(1) | 0.2210(65) | **0.5528(4)** |
| Core Beta Reweighted Proposal (0% clip) | 0.0065(1) | 0.2833(84) | 0.4592(50) |
| Flow Trained on *WT-MetaD* Dataset | | | |
| Heavy Atom Proposal | 0.0510(3) | 2.5527(382) | — |
| Heavy Atom Reweighted Proposal (1% clip) | 0.0063(1) | 0.2277(66) | 0.4115(4) |
| Heavy Atom Reweighted Proposal (0% clip) | 0.0058(1) | 0.2163(61) | 0.3603(23) |
| Core Beta Proposal | 0.0437(2) | 1.7185(294) | — |
| Core Beta Reweighted Proposal (1% clip) | 0.0057(1) | 0.2093(58) | 0.4818(4) |
| Core Beta Reweighted Proposal (0% clip) | 0.0058(1) | 0.2015(55) | 0.3939(11) |
| Atomistic BGs Trained on *Implicit Solvent* Dataset | | | |
| TarFlow | 0.0548(2) | 1.2696(314) | 0.0034(18) |
| ECNF++ | 0.1451(19) | 21.4005(4594) | 0.2445(156) |

*Table 8.* Quantitative comparison for alanine tripeptide across CG resolutions and different weight clipping thresholds.

| Model | JS (↓) | PMF (↓) | ESS (↑) |
|---|---|---|---|
| Flow Trained on *Unbiased* Dataset | | | |
| Heavy Atom Proposal | 0.0065(1) | 0.4071(73) | — |
| Heavy Atom Reweighted Proposal (1% clip) | **0.0056(1)** | **0.1957(52)** | 0.3201(4) |
| Heavy Atom Reweighted Proposal (0% clip) | 0.0110(1) | 0.2926(184) | 0.0368(192) |
| Core Beta Proposal | 0.0061(1) | 0.3800(71) | — |
| Core Beta Reweighted Proposal (1% clip) | 0.0060(1) | 0.2112(51) | **0.4212(5)** |
| Core Beta Reweighted Proposal (0% clip) | 0.0414(2) | 0.8451(142) | 0.0018(10) |

*Table 9.* Quantitative comparison for alanine hexapeptide across different weight clipping thresholds.

| Model | JS (↓) | PMF (↓) | ESS (↑) |
|---|---|---|---|
| Flow Trained on *Unbiased* Dataset | | | |
| Core Beta Proposal | 0.0134(1) | 0.9801(130) | — |
| Core Beta Reweighted Proposal (1% clip) | **0.0100(1)** | **0.3646(81)** | **0.1231(3)** |
| Core Beta Reweighted Proposal (0% clip) | 0.1972(3) | 8.0359(879) | 0.0002(3) |

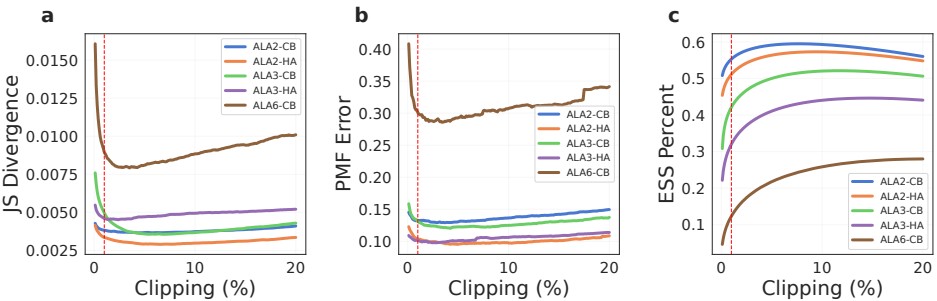

*Figure 10.* Metrics as a function of weight clipping ratio for flow models trained on unbiased datasets across different CG mappings and systems. (a) JS divergence, (b) PMF error, and (c) ESS after reweighting. The red dashed line indicates the 1% clipping ratio used throughout our experiments.

### G.5. Free Energy of $\psi$ Dihedral

We provide additional free energy plots for the $\psi$ dihedral of alanine dipeptide (Fig. 11).

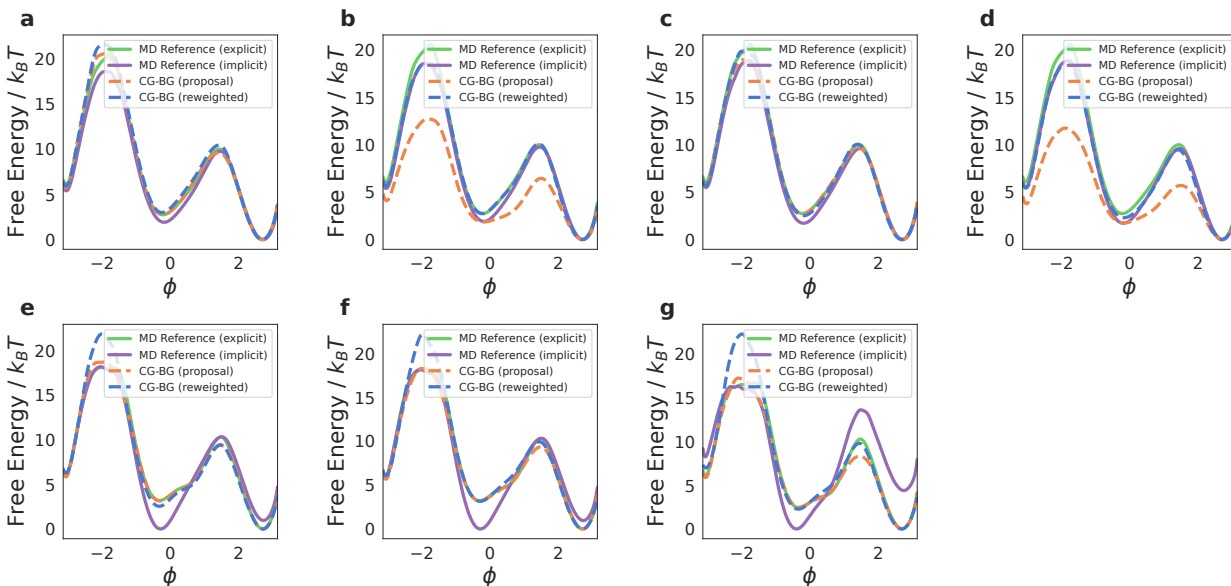

*Figure 11.* **Comparison of $\psi$ dihedral free energy profiles across training settings.** All panels show MD reference distributions and CG-BG proposals before and after reweighting. (a) Dipeptide Core Beta (unbiased), (b) Dipeptide Core Beta (WT-MetaD), (c) Dipeptide Heavy Atom (unbiased), (d) Dipeptide Heavy Atom (WT-MetaD), (e) Tripeptide Core Beta, (f) Tripeptide Heavy Atom, (g) Hexapeptide Core Beta.

### G.6. Bond Length

Fig. 12 shows the C–N bond length distributions for the MD reference and CG-BG results before and after reweighting.

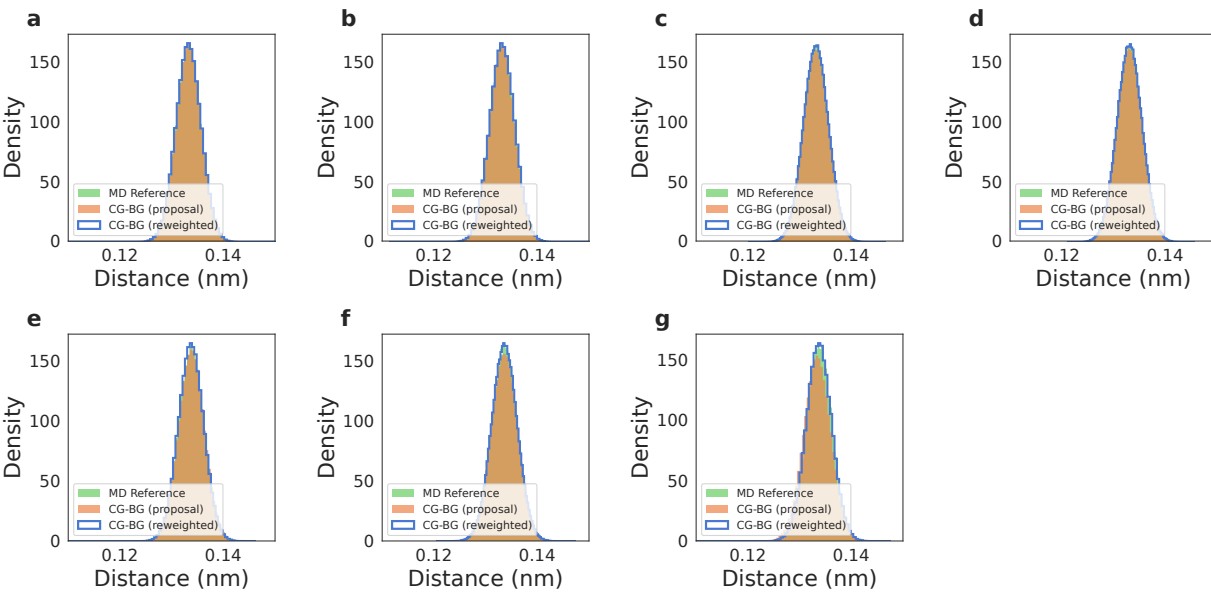

*Figure 12.* **Comparison of C–N bond length distributions across training settings.** All panels show MD reference distributions and CG-BG proposals before and after reweighting. (a) Dipeptide Core Beta (unbiased), (b) Dipeptide Core Beta (WT-MetaD), (c) Dipeptide Heavy Atom (unbiased), (d) Dipeptide Heavy Atom (WT-MetaD), (e) Tripeptide Core Beta, (f) Tripeptide Heavy Atom, (g) Hexapeptide Core Beta.

