# OpenReview forum: "Coarse-Grained Boltzmann Generators"
_ICML.cc/2026/Conference — ICML 2026 regular_

### Official Review · Reviewer_Xw8H · 2026-03-02

**Soundness:** 3
**Presentation:** 3
**Significance:** 3
**Originality:** 2
**Overall Recommendation:** 4
**Confidence:** 4

**Summary:**

The authors propose coarse-grained Boltzmann generators, which combine training a coarse-grained Boltzmann generator at the same time as a coarse-grained potential of mean force (PMF) for reweighting the coarse-grained Boltzmann generator proposal. They apply this methodology to sampling a toy system, the Müller-Brown potential, and alanine dipeptide in explicit solvent. Mainly driven by the reweighting capability through the learned PMF, the authors show good agreement of the reweighted distribution with the ground truth. The reweighted distribution further performs favorably compared to classical implicit solvent simulations.

**Compliance With Llm Reviewing Policy:**

Affirmed.

**Final Justification:**

The authors added additional systems that showcase that the method scales to larger systems. I am maintaining my already positive score.

**Key Questions For Authors:**

6. Given the proposed methodology, what do the authors think the application of it could look like in practice? Is this mainly useful in the transferable setting, where a transferable CG BG is combined with a transferable PMF? I think a discussion of the potential applicability in real-world settings would further strengthen the manuscript.

**Limitations:**

yes

**Strengths And Weaknesses:**

**Strengths**

- The authors address a key limitation of Boltzmann generators, which is the scalability to larger systems. Using a coarse-grained representation, Boltzmann generators can be trained in a lower-dimensional space, while still allowing reweighting using the learned PMF.
- The reweighted distribution shows good agreement with the provided ground truth data.
- The reweighted distribution outperforms a classical implicit solvent simulation.
- The authors provide an interesting ablation on the CG mapping choice for alanine dipeptide.

**Weaknesses**

1. The proposed approach is somewhat incremental in that it combines methods that are already known: training BGs in CG spaces and learning a PMF using force-matching on (biased) data. However, I have not seen the two combined in the way proposed in the given manuscript. However, while the contribution itself is somewhat incremental, the significance for further developments in this direction might be large.
2. The scalability of the shown approach has not been demonstrated by the authors. However, I believe the proposed methodology evaluated on alanine dipeptide might be strong and interesting enough on its own, and scalability can be tested in future work.

**Additional comments**

3. The authors state that their method enables “exact equilibrium sampling in a reduced-dimensional representation.” This is, however, only true if the PMF is learned exactly, which is basically never the case. Thus, even when reweighting using the PMF, a bias will always remain. This is in contrast to all-atom BGs, where exact reweighting guarantees are present, since one reweights directly toward the ground truth target. I think this fact can be phrased a bit better in the manuscript.
4. Metrics in Table 2 are only shown after reweighting. I think the benefit of reweighting with the PMF would be better-illustrated if metrics for the non-reweighted proposal were also included in the table.
5. I did not find a reproducibility statement in the current manuscript. Indicating that the source code and data used to reproduce the experiments will be made publicly available would strengthen the paper’s contribution to the field.

---

> ### Author Rebuttal · Authors · 2026-03-31
>
> We thank you for your positive assessment, and for recognizing the promise this framework holds for future developments in the field.
> We address the main concerns below.
> ### Q2. Scalability to Larger Systems
>
> We agree that demonstrating scalability is a crucial next step. We have added new experiments evaluating our method on larger systems, including tripeptide (3 residues)([plot_marginal_ala3](https://anonymous.4open.science/r/private-5325/ala3_dihedrals_marginal_CGBG_vs_MD.png), [plot_fes_ala3](https://anonymous.4open.science/r/private-5325/ala3_dihedrals_free_energy_CGBG_vs_MD.png)) and hexapeptide (6 residues)([plot_marginal_ala6](https://anonymous.4open.science/r/private-5325/ala6_dihedrals_marginal_CGBG_vs_MD.png), [plot_fes_ala6](https://anonymous.4open.science/r/private-5325/ala6_dihedrals_free_energy_CBGB_vs_MD.png)). These results with  [table_metrics](https://anonymous.4open.science/r/private-5325/ala3_and_ala6_metric_plus_w2_distance.md) demonstrate that our framework maintains strong performance and scalability as system size increases.
>
> ### Q3. The "Exact" Sampling Caveat
>
> We agree that because the PMF is machine-learned and thus an approximation of the true target energy, a bias will always remain in practice. We will revise the manuscript to clarify this nuance, ensuring we accurately state that we target the exact equilibrium formulation, but acknowledge the practical bias introduced by the PMF approximation. We will also expand on this in the Limitations section.
>
> ### Q4. Metrics Without Reweighting
>
> We have already included the metrics for the non-reweighted proposals in Table 6 in the Appendix (original submission). In the revised manuscript, we will add a more prominent pointer in the main text directing readers to this table so that the benefit of the reweighting step is immediately clear.
>
> ### Q5. Reproducibility
>
> We will add a clear reproducibility statement to the paper. The source code, datasets, and model weights are already prepared and will be made publicly open-source upon publication to ensure the community can build upon this work.
>
> ### Q6. Real-World Applications and Transferability
>
> We completely agree that transferability is the most promising direction for real-world applications. There is already strong evidence demonstrating the transferability of both generative models [1] and machine-learned PMFs [2].
>
> Advancing this direction will require a joint effort between the generative modeling and MLIP communities to find the optimal balance between the two paradigms. Specifically, while Boltzmann emulators can scale to larger systems, they inevitably produce some “hallucinated” or unphysical samples that the emulator alone cannot filter. Coupling the emulator with a learned PMF provides a necessary energy function to identify and remove these outliers.
>
> Furthermore, as our work demonstrates, CG MLPs can be trained using rapidly converged data. This efficiency could be instrumental in addressing the “circular dependency” problem often encountered in this field (i.e., the challenge of requiring unbiased data to train a model).
>
> Ultimately, for the current generation of atomistic Boltzmann Generators to be practically useful, it must move toward explicit-solvent accuracy. We believe that integrating a CG MLP is a likely necessary path forward to achieve this. We will gladly incorporate this broader perspective into the Discussion section of the revised manuscript.
>
> ---
>
> ### References
>
> [1] Lewis, S., Hempel, T., Jiménez-Luna, J., Gastegger, M., Xie, Y., Foong, A. Y., ... & Noé, F. (2025). *Scalable emulation of protein equilibrium ensembles with generative deep learning.*
>
> [2] Charron, N. E., Bonneau, K., Pasos-Trejo, A. S., Guljas, A., Chen, Y., Musil, F., ... & Clementi, C. (2025). *Navigating protein landscapes with a machine-learned transferable coarse-grained model.*

---

> > ### Author Rebuttal · Reviewer_Xw8H · 2026-04-02
> >
> > I thank the authors for their answers to my questions. The added systems now show that this also works for slightly larger systems. I am sticking to my original score because I believe it still adequately assesses the work.

---

> > > ### Author Response · Authors · 2026-04-07
> > >
> > > We thank the reviewer for the positive assessment and for their thoughtful feedback. We greatly appreciate your time and insights.

---

### Official Review · Reviewer_d9aE · 2026-03-12

**Soundness:** 2
**Presentation:** 3
**Significance:** 3
**Originality:** 3
**Overall Recommendation:** 4
**Confidence:** 4

**Summary:**

This paper introduces Coarse-Grained Boltzmann Generators, a framework designed to sample equilibrium molecular configurations efficiently. The method combines the potential of Boltzmann generators and importance sampling, with the scalability of coarse-grained emulators; by projecting the system into a lower-dimensional CG space and training two models in parallel: a flow-based generative model to propose CG configurations, and a machine learning Potential trained to act as the target energy for importance reweighting. This approach provides a computationally scalable way to achieve unbiased equilibrium sampling while implicitly capturing complex interactions.

**Compliance With Llm Reviewing Policy:**

Affirmed.

**Final Justification:**

The authors have addressed most of the concernes I raised, and therefore I have increased my score

**Key Questions For Authors:**

- As discussed, there are two main limitations in my opinion. The first one is the lack of experiments showing scalability. While I see the reasoning for why the authors argue this should scale, there are many reasons why it might not: Coarse graining means losing a lot of information, learning the CG potential could be complex as dimensionality increases, etc. The systems where the authors experiment, are systems where coarse graining is not necessary.

- The second limitation, is that even for those systems, no proper comparison with existing methods (FAB, iDEM, PITA and many others) is provided, which makes it very hard to know how well the method is actually doing. Comparing with those methods on the usually metrics (Wasserstein, ESS, FLD, PQMass, etc.) is required for serious evaluation of the results.

- It would be interesting to ablate over more choices of coarse graining.

**Limitations:**

Yes

**Strengths And Weaknesses:**

**Soundness**: The theoretical foundation is rigorous. The authors formally link the force matching objective to the Kullback-Leibler divergence (Proposition 1) and mathematically justify the use of biased data for learning the PMF (Propositions 2 and 3). The inclusion of ablations also helps. One of the mean weaknesses of the paper, however, are the experiments. The premise of the paper is that existing methods for boltzmann generation do not scale beyond toy problems, and things like alanine dipeptide. However, the authors only test their method on a toy problem, and alanine dipeptide! And, within those, there is no comparison to any state of the art alternative methods (FAB, PITA, iDEM, etc.).

**Presentation**: While the writing is generally good, and figure 1 is very helpful, the flow of the paper is strange. I was particularly confused to see a discussion of related work in section 5, right before the conclusions, which in my opinion repeats a lot of information that was already laid out in the introduction.

**Significance**: Sampling equilibrium conformations of molecular systems remains an important and unsolved problem. The idea of reducing the dimensionality of the space to sample is certainly interesting, as is the ability to work on implicit and explicit solvents.

**Originality**: While normalizing flows, force matching, and coarse-graining are all established concepts, unifying them into a single framework where a machine-learned PMF serves as the target energy for a Boltzmann Generator is an original idea.

---

> ### Author Rebuttal · Authors · 2026-03-31
>
> We thank the reviewer for the constructive feedback and for recognizing the novelty and potential impact of our framework. We address the main concerns below.
>
> ### Response to Weaknesses
>
> 1. **Scalability and the role of coarse-graining**
>    While coarse-graining involves information loss and PMF learning grows complex in higher dimensions, existing literatures ([1, 2]) demonstrates that coarse-grained machine learning potentials scale successfully. Coarse-graining does not simply mean loss of information; it also provides the flexibility to bias coarse-grained degrees of freedom to accelerate the data generation process, as demonstrated in our manuscript. Most importantly, atomistic BGs fundamentally cannot sample peptides in explicit solvent systems.
>
>    To directly address scalability, we added new experiments on larger systems: tripeptide (3 residues)([plot_marginal_ala3](https://anonymous.4open.science/r/private-5325/ala3_dihedrals_marginal_CGBG_vs_MD.png), [plot_fes_ala3](https://anonymous.4open.science/r/private-5325/ala3_dihedrals_free_energy_CGBG_vs_MD.png)) and hexapeptide (6 residues)([plot_marginal_ala6](https://anonymous.4open.science/r/private-5325/ala6_dihedrals_marginal_CGBG_vs_MD.png), [plot_fes_ala6](https://anonymous.4open.science/r/private-5325/ala6_dihedrals_free_energy_CBGB_vs_MD.png)) . These results show our method scales comparably to SOTA atomistic BGs while achieving higher accuracy. We also provided [table_metrics](https://anonymous.4open.science/r/private-5325/ala3_and_ala6_metric_plus_w2_distance.md) for these larger systems including Wasserstein distance for further comparison.
>
> 2. **Organization and related work.**
>    In the revised manuscript, we will rewrite the Related Work section to strictly outline the technical differences between our approach and previous methods, avoiding repetition of established concepts already covered in the Introduction.
>
> ---
>
> ### Responses to Questions
>
> **Q2. Comparison with existing methods (FAB, iDEM, PITA, etc.).**
> We appreciate the suggestion to compare against these methods. However, FAB, iDEM, and PITA are "data-free" neural samplers designed to sample unnormalized densities by directly querying energy functions, without access to samples from MD simulation. This is conceptually different from flow models trained with samples. Because the underlying setups differ, a direct empirical comparison is not an apples-to-apples evaluation. That said, replacing the implicit force fields typically used in these data-free neural samplers with our learned PMF could increase their scalability and accuracy, which is a promising future direction.
>
> A primary reason for the initial lack of comparisons with other work is that atomistic BGs use implicit solvent MD as their ground truth. Our target accuracy is fundamentally higher than their baselines. To provide a clearer evaluation, we have added new experiments comparing our framework to recent SOTA atomistic BG models, including ECNF++ and TarFlow [3]. As expected, their accuracy ([metrics](https://anonymous.4open.science/r/private-5325/ala2_metrics_compare_with_other_model_plus_w2_distance.md)) falls behind the implicit solvent baselines in this explicit-solvent context.
>
> **Q3. Ablation over coarse-graining choices.**
> We agree that it is interesting to explore different CG mappings. While normalizing flows place no restrictions on the mapping used, the *PMF learning* is highly sensitive to this choice. As demonstrated in recent literature [4], not all mappings are viable for learning a reliable PMF. Therefore, we intentionally restricted our ablations to mappings that are theoretically and empirically grounded for PMF learning, rather than testing arbitrary choices. We will leave the ablation of larger new added systems for future work.
>
> ---
>
> ### References
>
> [1] Charron, N. E., Bonneau, K., Pasos-Trejo, A. S., Guljas, A., Chen, Y., Musil, F., ... & Clementi, C. (2025). *Navigating protein landscapes with a machine-learned transferable coarse-grained model.*
>
> [2] Chen, Y., Krämer, A., Charron, N. E., Husic, B. E., Clementi, C., & Noé, F. (2021). *Machine learning implicit solvation for molecular dynamics.*
>
> [3] Tan, C. B., Hassan, M., Klein, L., Syed, S., Beaini, D., Bronstein, M. M., ... & Neklyudov, K. (2025). *Amortized sampling with transferable normalizing flows.*
>
> [4] Görlich, F., & Zavadlav, J. (2026). *Mapping Still Matters: Coarse-Graining with Machine Learning Potentials.*

---

> > ### Author Rebuttal · Reviewer_d9aE · 2026-04-07
> >
> > I appreciate the rebuttal from the authors. The addition of the tripeptide and hexapeptide experiments does show scalability to higher dimensions. Drawing the distinction with data-free" neural samplers is also a fair point. I would still have liked to see an ablation over coarse-graining choices, but my concerns have otherwise been addressed, therefore I am raising my score.

---

### Official Review · Reviewer_ZamS · 2026-03-12

**Soundness:** 3
**Presentation:** 2
**Significance:** 3
**Originality:** 3
**Overall Recommendation:** 4
**Confidence:** 4

**Summary:**

The paper proposes Coarse-Grained Boltzmann Generators, which combine scalable reduced-order models with an importance sampling stage to perform Boltzmann sampling of atomistic systems. The approach is tested on the Müller-Brown potential and alanine dipeptide using explicit solvent simulations to demonstrate the efficacy and scalability of the approach.

**Compliance With Llm Reviewing Policy:**

Affirmed.

**Final Justification:**

The authors addressed my concerns regarding improved benchmarking and provided far more clarity around the performance of their proposed approach against state-of-the-art.

**Key Questions For Authors:**

- How accurately does the learned PMF approximate the true marginal distribution?
- Is the learned model invertible? Can this be proven/shown in this context?
- Can other global and local metrics for measuring performance be included, e.g., 2-Wasserstein distances on the energy distribution and the dihedral angles?
- In the implicit solvent setting, how well does the proposed approach perform relative to the state-of-the-art all-atom BGs for peptide sampling (https://arxiv.org/abs/2508.18175; https://arxiv.org/abs/2512.09914)?
- Can the system be scaled to larger molecules like tetrapeptides or octapeptides to demonstrate the scalability of the method?

**Limitations:**

Yes

**Strengths And Weaknesses:**

**Strengths**

The idea is quite interesting in that the objective is to use coarse-graining and potentials of mean force to lower the dimensionality of the atomistic system while, in principle, preserving an exact likelihood model that can be used in conjunction with importance sampling for de-biasing samples and generating conformations of peptides/proteins of interest. The theoretical objective is well-motivated, clearly presented, and of practical relevance and interest.

The propositions included are relevant and support the importance of accurately capturing the marginal density prior to performing self-normalized importance sampling as a re-weighting step. Further, the empirical performance on the demonstrated tasks is quite strong.

**Weaknesses**

Although the work is theoretically interesting, the practical implementation leaves several important questions unresolved. In principle, coarse-graining combined with the PMF defines an exact marginal equilibrium distribution over the coarse variables. This statement holds only if the PMF is computed exactly, which is generally intractable in practice. As a result, the validity of the resulting probability measure depends heavily on how accurately the PMF is approximated by the learned model. The paper does not clearly demonstrate that the learned PMF provides a sufficiently accurate approximation of this marginal distribution.

The authors propose incorporating an importance sampling stage to re-weight samples and correct biases in generated configurations. While this idea is reasonable in principle, its effectiveness depends on the quality of the proposal distribution and the validity of the likelihoods used for re-weighting. In particular:

- Importance sampling requires reliable likelihood ratios with respect to the atomistic Boltzmann distribution.
- If the model does not provide a consistent density or an accurate estimator of these likelihood ratios, the resulting importance weights may be unreliable.
- Improvements in downstream metrics after re-weighting therefore do not necessarily imply that the underlying likelihood model is correct or that the generated samples faithfully represent the true atomistic distribution.

In practice, the success of this approach depends critically on the fidelity of the learned PMF. The paper does not convincingly establish that the learned PMF is sufficiently accurate for the proposed workflow.

The introduction also states that prior Boltzmann Generators have struggled to scale beyond short peptides. While this characterization may apply to earlier work, the paper does not adequately address more recent developments. In particular, the authors cite recent work (https://arxiv.org/abs/2508.18175) demonstrating that discrete-time normalizing flows can scale to larger systems and achieve high-quality zero-shot generation of unseen octapeptides by conditioning on sequence. Other works have also clearly demonstrated the ability to scale to peptides of sequence length six or more (https://arxiv.org/abs/2512.09914). Although these works operate under implicit solvent models—which the authors mention—none of these methods are used as a baseline for comparison in the implicit solvent setting.

The claim that Boltzmann Generators struggle to scale would be more convincing if the proposed method clearly demonstrated improved scalability. The experiments, however, are limited to the Müller–Brown potential and alanine dipeptide. The latter is among the smallest benchmark systems used in prior literature. While such systems are commonly used for proof-of-concept studies, excluding stronger baselines while suggesting that the method enables scaling to larger systems risks overstating the contribution. As presented, the empirical results do not convincingly support the claim that the approach provides a practical path toward scaling Boltzmann Generators to larger molecular systems.

---

> ### Author Rebuttal · Authors · 2026-03-31
>
> We thank the reviewer for the thoughtful feedback and for recognizing the promise of our work. We address the main concerns below.
>
> ### Response to Weaknesses
>
> 1. **Accuracy of the learned PMF.**
>    We agree that the validity of our framework depends critically on how well the learned PMF approximates the true marginal distribution. In the original submission, Figure 5 provides indirect but meaningful evidence: after self-normalized importance reweighting, the generated samples match the atomistic reference distribution closely. This outcome is non-trivial; if the learned PMF were inaccurate, the reweighted distribution would not match the target.
>
>    To address this concern more directly, we have added additional results ([plot_marginal](https://anonymous.4open.science/r/private-5325/ala2_dihedrals_marginal_learned_PMF_vs_MD.png) and [plot_rama](https://anonymous.4open.science/r/private-5325/ala2_rama_learned_PMF_vs_explicit_solvent.png)), where we perform long NVT MD simulations using the learned PMF and compare the resulting statistics against atomistic references, demonstrating that the learned PMF is sufficiently accurate, much better than implicit solvent MD.
>
>    Additionally, our PMF setup follows prior work [1], where strong agreement between learned and reference PMFs has been demonstrated.
>
> 2. **Scalability and comparison to prior work.**
>    We appreciate the reviewer highlighting recent advances in scaling Boltzmann Generators. We would like to clarify that our approach is complementary rather than directly competitive with these methods. While recent works demonstrate scaling to octapeptides under implicit solvent models [2, 3], these are still short peptides ($\leq 10$ residues), and their accuracy is fundamentally limited by the implicit solvent approximation.
>
>    In contrast, we focus on achieving *explicit-solvent-level accuracy* through coarse-graining and PMF modeling. Importantly, our method is architecture-agnostic and can incorporate state-of-the-art atomistic BGs as the flow backbone or autoregressive backbone [4], allowing it to naturally benefit from recent advances in scalable architectures. In other words, any progress in exact-likelihood generative architectures can also benefit our method.
>
>    To further demonstrate scalability, we have added new experiments on larger systems such as tripeptide (3 residues)([plot_marginal_ala3](https://anonymous.4open.science/r/private-5325/ala3_dihedrals_marginal_CGBG_vs_MD.png), [plot_fes_ala3](https://anonymous.4open.science/r/private-5325/ala3_dihedrals_free_energy_CGBG_vs_MD.png)) and hexapeptide (6 residues)([plot_marginal_ala6](https://anonymous.4open.science/r/private-5325/ala6_dihedrals_marginal_CGBG_vs_MD.png), [plot_fes_ala6](https://anonymous.4open.science/r/private-5325/ala6_dihedrals_free_energy_CBGB_vs_MD.png)) . For these larger systems, our approach achieves improved agreement with atomistic references compared to implicit solvent baselines ([table_metrics](https://anonymous.4open.science/r/private-5325/ala3_and_ala6_metric_plus_w2_distance.md)), thus outperforming all current atomistic BGs for these specific systems.
>
>
>
> ---
>
> ### Responses to Specific Questions
>
> **Q2.** Yes. The model is based on normalizing flows, which are bijective by construction.
>
> **Q3.** We have added additional evaluation ([metrics](https://anonymous.4open.science/r/private-5325/ala2_metrics_compare_with_other_model_plus_w2_distance.md)), including Wasserstein distances on dihedral angle distributions, to provide a more comprehensive assessment.
>
> **Q4.** We further clarify that our ground truth is explicit solvent accuracy, which current state-of-the-art atomistic BGs cannot achieve. To demonstrate this, we additionally show results from SOTA BGs such as TarFlow and ECNF++ ([Table](https://anonymous.4open.science/r/private-5325/ala2_metrics_compare_with_other_model_plus_w2_distance.md)), which, as expected, exhibit larger errors than implicit solvent MD baselines. To better illustrate the magnitude of the difference between implicit and explicit solvents, we also provide the marginal dihedral plots for tripeptide ([plot_marginal_ala3](https://anonymous.4open.science/r/private-5325/ala3_dihedrals_marginal_CGBG_vs_MD.png)) and hexapeptide ([plot_marginal_ala6](https://anonymous.4open.science/r/private-5325/ala6_dihedrals_marginal_CGBG_vs_MD.png)).
>
> ---
> ### References
>
> [1] Chen, W., Görlich, F., Fuchs, P., & Zavadlav, J. (2025). *Enhanced Sampling for Efficient Learning of Coarse-Grained Machine Learning Potentials.*
>
> [2] Rehman, D., Akhound-Sadegh, T., Gazizov, A., Bengio, Y., & Tong, A. (2025). *FALCON: Few-step Accurate Likelihoods for Continuous Flows.*
>
> [3] Tan, C. B., Hassan, M., Klein, L., Syed, S., Beaini, D., Bronstein, M. M., ... & Neklyudov, K. (2025). *Amortized sampling with transferable normalizing flows.*
>
> [4] Rehman, D., Tan, C. B., Bengio, Y., Bose, J., & Tong, A. (2026). *Autoregressive Boltzmann Generators.*

---

> > ### Author Rebuttal · Reviewer_ZamS · 2026-04-03
> >
> > I would like to thank the authors for their clarifications and answering my raised concerns. All of them have been addressed, and as a result, I'm increasing my score.

---

> > > ### Author Response · Authors · 2026-04-07
> > >
> > > We thank the reviewer for their careful evaluation and the time spent on our rebuttal. We are glad that the additional experiments and extended comparisons addressed your concerns and contributed to an increased score.

---

### Decision · Program_Chairs · 2026-04-30

**Decision:**

Accept (regular)

**Comment:**

This paper studies the problem of sampling from the Boltzmann distribution of a molecule. The paper augments the Boltzmann generators approach with coarse-grained surrogates, with the goal of scaling the sampling process. During the initial review, all reviewers generally found the idea to be interesting, well motivated, and of practical interest to the community.

Some of the weaknesses identified across reviews include:
* One of the main claims of the work is to address scalability, but the experiments themselves don't demonstrate scalability
* The method itself may be incremental in that it combines well-studied ideas

During the rebuttal, many new experiments were completed to address the scalability criticisms, and several reviewers increased their scores to weak accept as a result. Despite the criticism of the paper being incremental, reviewers thought that it still showed promise. Thus, I recommend accepting the paper based on reviewer consensus.